

# International Monitoring System infrasound data products for atmospheric studies and civilian applications

Patrick Hupe[1], Lars Ceranna[1], Alexis Le Pichon[2], Robin S. Matoza[3], Pierrick Mialle[4]

[1] BGR, B4.3, D-30655 Hannover, Germany
[2] CEA, DAM, DIF, F-91297 Arpajon, France
[3] Department of Earth Science and Earth Research Institute, University of California, Santa Barbara, CA, USA
[4] CTBTO, IDC, Vienna, Austria

*Correspondence to*: Patrick Hupe (Patrick.Hupe@bgr.de)

## Abstract

The International Monitoring System (IMS) has been established since the late 1990s for the verification of the Comprehensive Nuclear-Test-Ban Treaty (CTBT). The IMS is supposed to detect any explosion of at least 1 kt of TNT equivalent underground, underwater, and in the atmosphere. Upon completion, monitoring the Earth's atmosphere for low-frequency pressure waves will be realized using up to 60 infrasound stations distributed over the globe. Acoustic waves in the infrasound range (between around 0.01 and 20 Hz) can efficiently propagate over long distances, subject to the winds near the stratopause at around 50

km. Therefore, infrasound observations of repeating or persistent sources have been suggested for probing the winds in the middle atmosphere, where numerical weather prediction models suffer from the lack of continuous observation technologies for data assimilation. One type of repetitive source is active volcanoes. In turn, this natural hazard for civil security can be monitored using infrasound, and first prototypes of applications for the release of early volcanic eruption warnings have been established. However, access to raw infrasound data or products of the IMS is limited to specific user groups, which might

hinder the utilization of infrasound observations.

In this study, we present advanced infrasound data products for atmospheric studies and civilian applications. For this purpose, 18 years of raw infrasound data (2003-2020) were reprocessed using the Progressive Multi-Channel Correlation method. A one-third octave frequency band configuration between 0.01 and 4 Hz was chosen for running this array-processing algorithm, which detects coherent infrasound waves within the background noise. From the comprehensive detection lists, each four

products for 53 IMS infrasound stations were derived. The four products cover different frequency ranges and are provided at different temporal resolutions: a very low frequency set (0.02-0.07 Hz, 30 min; https://doi.org/10.25928/bgrseis_bblf-ifsd, Hupe et al., 2021a), two so-called microbarom frequency sets – covering both the lower (0.15-0.35 Hz, 15 min; https://doi.org/10.25928/bgrseis_mblf-ifsd, Hupe et al., 2021b) and a higher (0.45-0.65 Hz, 15 min; https://doi.org/10.25928/bgrseis_mbhf-ifsd, Hupe et al., 2021c) part – named after the dominant ambient noise of interacting

ocean waves that is quasi-continuously detected at IMS stations, and observations with center frequencies of 1 to 3 Hz (5 min), called the high frequency product (https://doi.org/10.25928/bgrseis_bbhf-ifsd, Hupe et al., 2021d). Within these frequency





ranges and time windows, the signals from the most dominant directions in terms of number of arrivals are summarized. Along with several detection parameters, calculated quantities for assessing the relative quality of the products are provided. The validity of the data products is demonstrated by diving into examples of recent events that produced infrasound detected at
IMS infrasound stations, as well as a global assessment.

## 1 Introduction

After the Comprehensive Nuclear-Test-Ban Treaty (CTBT) was opened for signature in 1996, the International Monitoring System (IMS) has been established for monitoring compliance with the treaty (Dahlmann et al., 2009). When completed, this monitoring and verification infrastructure will consist of 337 facilities, composing of 170 seismic, 11 hydro-acoustic, and 60
infrasound stations for detecting clandestine nuclear tests underground, underwater, and in the atmosphere, respectively, as well as 80 radionuclide detectors and 16 laboratories for providing evidence of the nuclear character of an explosion (e.g., Marty, 2019). Both the waveform and radionuclide technologies record the data continuously.

Infrasound defines pressure fluctuations in a range between the acoustic cut-off frequency (3-10 mHz) and the lower human-hearing frequency threshold of sound (16-20 Hz). At low infrasonic frequencies, acoustic waves can travel long distances over
the atmosphere, ranging from hundreds to several thousands of kilometers (e.g., De Groot-Hedlin et al., 2010), subject to the dynamics in the different atmosphere layers (Drob et al., 2003). Propagation distances of more than 1,000 km generally require a waveguide below the stratopause at around 50 km, as the attenuation loss significantly increases in the mesosphere (~50-90 km) and particularly the thermosphere above 90 km (e.g., Sutherland and Bass, 2004). A waveguide establishes when the effective sound speed (this adds the along-path wind speed to the speed of sound) at an altitude exceeds the speed of sound at
the surface such that upward-propagating acoustic waves are refracted downward – the so-called effective sound speed ratio ($v_{\text{eff-ratio}}$) exceeds one (e.g., Wilson, 2003; Le Pichon et al., 2012). Along with the temperature maximum near the stratopause, the direction of the strong stratospheric winds is decisive of whether a stable surface-to-stratosphere waveguide establishes. Ducting in the troposphere (<15 km) or between the surface and the thermosphere generally constrains the propagation ranges to a few hundred up to around one thousand kilometers (Drob et al., 2003), although exceptions such as the June 2009 Sarychev
Peak eruption (Matoza et al., 2011a) and the very low frequency mountain-associated waves (MAWs; Hupe et al., 2019b) were found.

Due to the efficient low-frequency ducting and highly sensitive pressure sensors – microbarometers –, IMS infrasound stations are capable of recording small pressure fluctuations of a few millipascals, which can originate from remote infrasound sources. At each IMS infrasound station, at least four microbarometers with a flat response from 0.02 Hz to 4 Hz are arranged to an
array of an aperture between one and three kilometers, thus functioning as an acoustic antenna. Cross-correlation methods enable deducing properties of waves passing the array, such as the incoming direction (back azimuth). If signatures of a source are detected at two or more stations, the origin time, location, and potentially other source characteristics can be determined. The design goal of the IMS infrasound network targets at detecting and locating any explosion in the atmosphere with a yield





of at least one kiloton trinitrotoluene (TNT; $1\,\text{kt}$ of $\text{TNT} = 4.185 \times 10^{12}$ J) equivalent (Christie and Campus, 2010).

Consequently, besides the latest (underground) nuclear tests (e.g., Assink et al., 2016; Koch and Pilger, 2019), also accidental explosions are detected by the infrasound stations (e.g., Ceranna et al., 2009; Green et al. 2011; Pilger et al., 2021a). Moreover, a variety of natural sources is captured by the sensors, including large meteorites entering the Earth's atmosphere (e.g., Arrowsmith et al., 2008; Le Pichon et al., 2013; Pilger et al., 2020), volcanic eruptions (e.g., Campus, 2006; Dabrowa et al., 2011; Matoza et al., 2013, 2019; Marchetti et al., 2019), and microbaroms from the oceans (e.g., Landès et al., 2014; De Carlo

et al., 2021).

It has been demonstrated that infrasonic signatures, especially those originating from repeating or quasi-continuous sources, can be used for probing the atmosphere and assessing atmospheric models (Le Pichon et al., 2009, 2019a; Blanc et al., 2019). For instance, Smets and Evers (2014) studied the life cycle of a sudden stratospheric warming (SSW) event using microbarom observations and showed that the microbaroms' amplitude variations allow deducing the propagation waveguide. The authors

concluded that the state of the atmosphere was well represented by the high-resolution (HRES) operational atmospheric model analysis of the European Centre for Medium-Range Weather Forecasts (ECMWF), but they also found some discrepancies related to the SSW event. Hupe et al. (2019a) considered seven months of temperature profiles obtained by collocating lidar to a German infrasound station for perturbing the wind and temperature of the ECMWF model and comparing the resulting microbarom simulations with observations. They showed that the model perturbations around the stratopause enabled a better

explanation of the microbarom detections and their amplitude variation during the summer. Also including lidar, Le Pichon et al. (2015) analyzed different ground-based and space-borne observation technologies, revealing systematic biases for temperature and wind in both analysis and reanalysis models. The authors concluded that such biases are critical to propagation simulations in the context of the CTBT verification. In turn, infrasound has been used to probe the middle atmosphere winds and crosswind effects along the propagation paths, based on active volcanoes as another repetitive source (Le Pichon et al.,

85  2005).

Therefore, several of those studies concluded that infrasound measurements can be used as a passive remote sensing technique for supplementing observational data of the middle atmosphere. Infrasound therefore has the potential to be assimilated in weather or climate models, which lack of observations at those altitudes (Le Pichon et al., 2015; Blanc et al., 2018). As a demonstrator, Amezcua et al. (2020) performed offline assimilation tests using infrasound data from 370 ground-truth

explosion events in Scandinavia and wind data of ECMWF's ERA5 model. They determined the largest impact of the assimilation in the stratosphere.

However, a hurdle for further exploring the potential of infrasound for enhancing the representation of winds in numerical weather prediction and analysis models is imposed by the fact that access to the IMS infrasound data is restricted. The raw waveform data are available to the National Data Centers (NDCs) of the CTBT, and data products such as the Reviewed Event

Bulletin (REB) of the CTBT Organization's (CTBTO) International Data Centre (IDC) in Vienna are accessible by nominated users. Limited data access can be granted to third parties and researchers through the virtual Data Exploitation Centre (vDEC) of the IDC.

The objective of this study is to provide products derived from detection lists of all IMS infrasound stations that can serve as observational data for the atmospheric community as well as for other scientific and civilian applications. A focus is on the frequency range of microbaroms (0.1-0.6 Hz), as these are a quasi-continuous, coherent ambient noise source at the majority of infrasound stations and thus convenient to probe the stratospheric circulation (e.g., Landès et al., 2012; Assink et al., 2014; Ceranna et al., 2019). We additionally provide a low-frequency product (0.02–0.07 Hz), which covers phenomena such as MAWs (e.g., Hupe et al., 2019b) and aurora infrasound (e.g., Wilson et al., 2010), as well as a high-frequency product (1–3 Hz) particularly covering surf and transient events. The latter product also includes signatures of volcanic eruptions, while volcano infrasound can generally feature a broad frequency range including the very low frequencies and the microbarom range (e.g., Matoza et al., 2019).

The chosen frequency ranges of the data products are a result of the comprehensive processing of all IMS infrasound data from 2003 to 2020. Details about the IMS data, the processing method, and the processing configuration are given in Section 2. Section 3 deals with the detection lists summarizing the processing results, which have already been utilized to validate a microbarom model (De Carlo et al., 2021) and to identify signatures of rocket launches for space missions (Pilger et al., 2021b). The products are elaborated in Section 4, including descriptions of the parameters that are part of the final data sets and thus relevant to users. In Section 5, we assess the data products based on selected explosive events and a global comparison of coherent ambient noise detections. Details on the data availability and access are described in Section 6.

## 2 Data and methods

### 2.1 IMS infrasound network

The infrasound network of the IMS will consist of 60 globally distributed stations, of which the CTBTO Preparatory Commission has certified 53 by the end of 2020 (CTBTO Preparatory Commission, 2020), shown in Fig. 1a. The remaining stations are planned, under construction, or even already installed but not certified. Table A1 in the Appendix A provides details on the location and geometry of the 53 certified arrays. Although the infrasound part of the IMS has not yet been completed and the speed of certifications has languished (Fig. 1b) due to logistic, security, or political reasons (Marty, 2019), studies investigating the detection capability have shown that the infrasound network already meets the design goal of the IMS (e.g., Le Pichon et al., 2019b).

IMS infrasound arrays consist of at least four sensors. Each sensor is equipped with a wind noise reduction system (WNRS); for instance, a pipe array connecting several air inlets serves as spatial filter for reducing the effect of small-scale disturbances at a sensor (e.g., Marty, 2019). Such WNRS application is particularly efficient at frequencies above 0.5 Hz for enhancing the signal-to-noise ratio (SNR) in the differential pressure records (Alcoverro and Le Pichon, 2005). The IMS microbarometers continuously sample the differential pressure at a rate of 20 Hz. For a four-element array, which applies to 15 of the 53 certified stations, the raw data amount to more than 6.9 million samples per day; for 23 stations, this amount is even doubled (eight elements). The remainder of the arrays comprises of between five and ten sensors, with one exception being IS23 on Kerguelen


Islands in the southern Indian Ocean (15 sensors). As 53 IMS arrays are certified, a data availability of 100 % would imply

that 357 microbarometers provided raw data (>615 million samples per day). Despite strict data availability requirements being

in place for the IMS network, single sensors can be temporarily down because of environmental conditions, equipment failure,

or maintenance measures (e.g., Marty, 2019). Station upgrades also lead to lacks of data since these often require a station to

be revalidated. Such upgrading activities, which are not depicted in Fig. 1b, include the installation of additional sensors, the

replacement of aged microbarometers, or the relocation of sensors to enhance the array response. In a few cases, even the

entire array was relocated; for instance, this happened in 2009 when IS27 in Antarctica was moved almost 5 km southward to

the new Neumayer III station on the Ekström Ice Shelf.

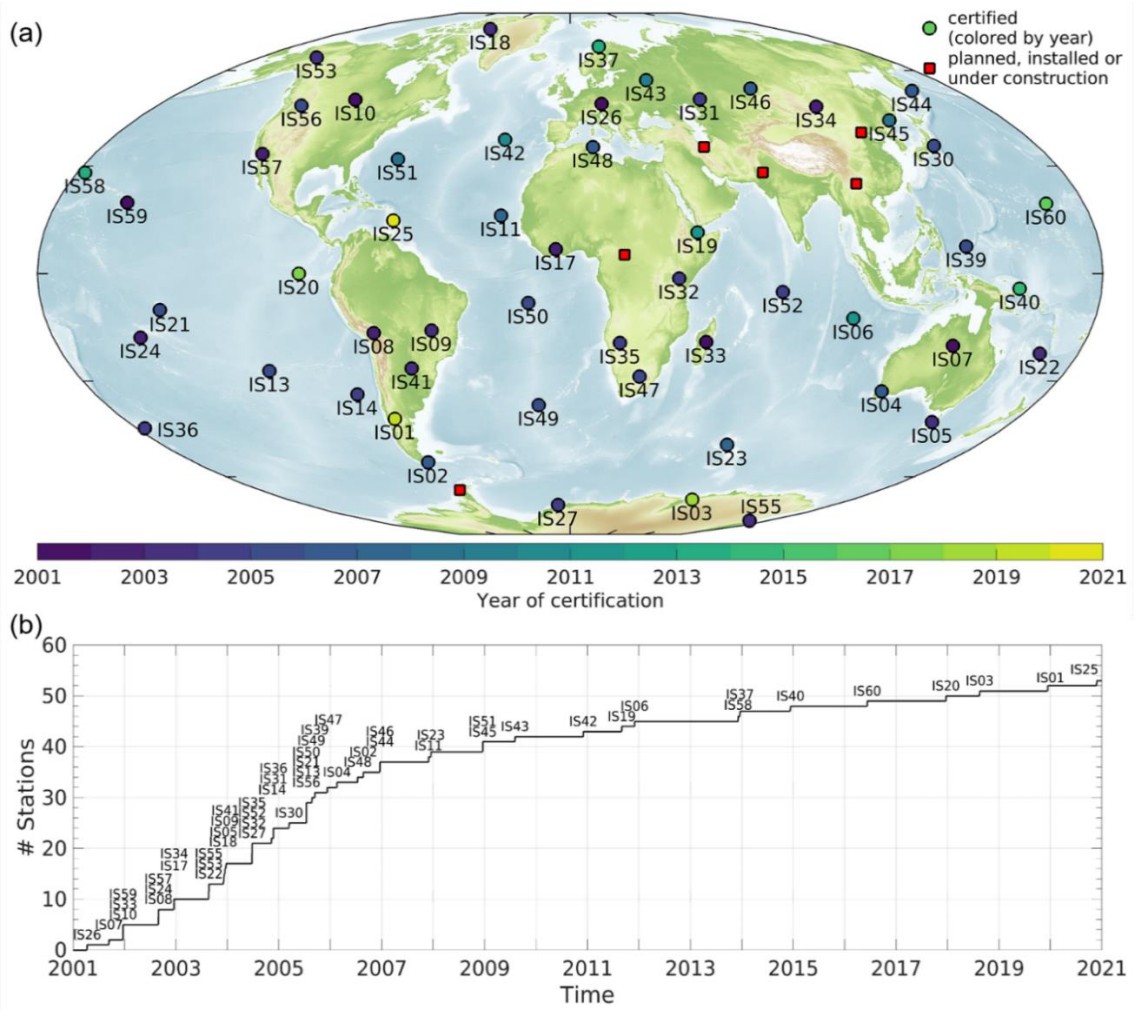

**Figure 1: (a) Map of the IMS infrasound network with certified and planned stations (as of January 2021). This study covers the**
**period from 2003 to 2020, during which the number of stations in operation was subsequently increased, as depicted by the**
**approximate certification timeline (b). All certified stations are considered in this study, although the detection lists and products of**
**IS25 (yellow circle) are not too expressive because of the short period the array was certified (end of 2020). The location of one of**
**the 60 stations has not been determined yet. The map is based on the ETOPO1 Global Relief Model (NOAA National Geophysical**
**Data Center, 2009; see also Amante and Eakins, 2009).**





The temporal loss of sensor data diminishes a station's detection capability, which can also reduce the detection capability of the entire network, especially if a station consisting of only four or five sensors is affected. Extreme events such as wild fires, flooding, or lightning strikes are potential causes of data losses. Moreover, the station-specific environment creates a spatiotemporal variation in the detection capability, including incoherent wind noise and unwanted coherent infrasonic signals (e.g., Ceranna et al., 2019). While wind noise raises the detection threshold for coherent signals, unwanted but coherent signals

may interfere the detection or discrimination of signals of interest. Therefore, the performance of the IMS stations is a key concern for the CTBT. Matoza et al. (2013) conducted a first comprehensive and systematic broadband analysis of historical IMS infrasound data and derived both station-specific coherent noise levels and overall performance characteristics. Ceranna et al. (2019) extended their broadband analysis by four years, thus covering the period from 01 April 2005 to 01 January 2015. For this study, we reprocessed all historical IMS infrasound data from 01 January 2003 to 31 December 2020 using an advanced

processing configuration.

## 2.2 Data processing

For automatically processing continuous waveform data of all infrasound stations, the Progressive Multi-Channel Correlation (PMCC) method (Cansi, 1995; Le Pichon and Cansi, 2003) is utilized. This array-processing algorithm proved to be efficient for detecting coherent low-amplitude acoustic waves within incoherent noise as PMCC is routinely used for processing the

IMS infrasound data at the IDC (Mialle et al., 2019). PMCC analyses the waveform data in successive, overlapping time windows and predefined frequency bands. The broadband analyses of Matoza et al. (2013) and Ceranna et al. (2019) based on a PMCC implementation with variable window lengths and 15 logarithmically spaced frequency bands (Fig. 2a). Brachet et al. (2010) pointed out that such implementations enable to reprocess the full frequency range (0.01–5 Hz) in a single computational run and thus outperform the initial implementation that relied on linearly spaced frequency bands. An enhanced

discrimination between interfering signals is envisaged through the use of one-third octave frequency bands (Garcés, 2013). We pursue such an enhancement and apply the one-third octave frequency bands that are depicted in Fig. 2b, with 2nd order Chebyshev filters. The 26 bands cover the frequency range from 0.01 Hz to around 4 Hz. The window lengths decrease with frequency from 600 s to about 23 s. The time step is 10 % of the respective window length (90 % overlap).

The wave front parameters of coherent plane waves are derived from cross-correlation functions and the arrival time delays

between sensor triplets of an array (Cansi, 1995); therefore at least three sensors need to be available. The more sensors are progressively incorporated (generally from the inside to the outside of an array), the more potential aliasing is limited, hence the better is the precision of the estimated parameters (Cansi and Le Pichon, 2008). Sub-networks (here triplets of array elements) can be predefined for the processing. It is of note that, beyond the minimum requirements for IMS infrasound arrays, not only the number of elements and the apertures but also the array geometries differ throughout the IMS (e.g., Marty, 2019),

limiting the comparability of processing results between stations. For the same reasons, the sub-networks cannot be selected uniformly. However, as the frequency range of the processing reaches down to 0.01 Hz and thus longer periods and acoustic wavelengths, the sub-network geometries are generally chosen to exploit the maximum array element separations at each array





(Matoza et al., 2013). Apart from that, mixing of "L"-type and "H"-type sensors within a sub-network – the letter indicates the applied WNRS aperture – is avoided. Meanwhile, stations formerly equipped with mixed sensor types have been subsequently

updated and homogenized. Apart from three exceptions where five sub-networks have proved to be efficient, we generally select four sub-networks for each array, and thus follow the previous processing configuration by Ceranna et al. (2019). PMCC is supposed to reconfigure the sub-networks automatically in the case of a sensor failure (i.e, lack of data). Table A1 in Appendix A contains two columns with the array apertures and the chosen sub-networks, respectively.

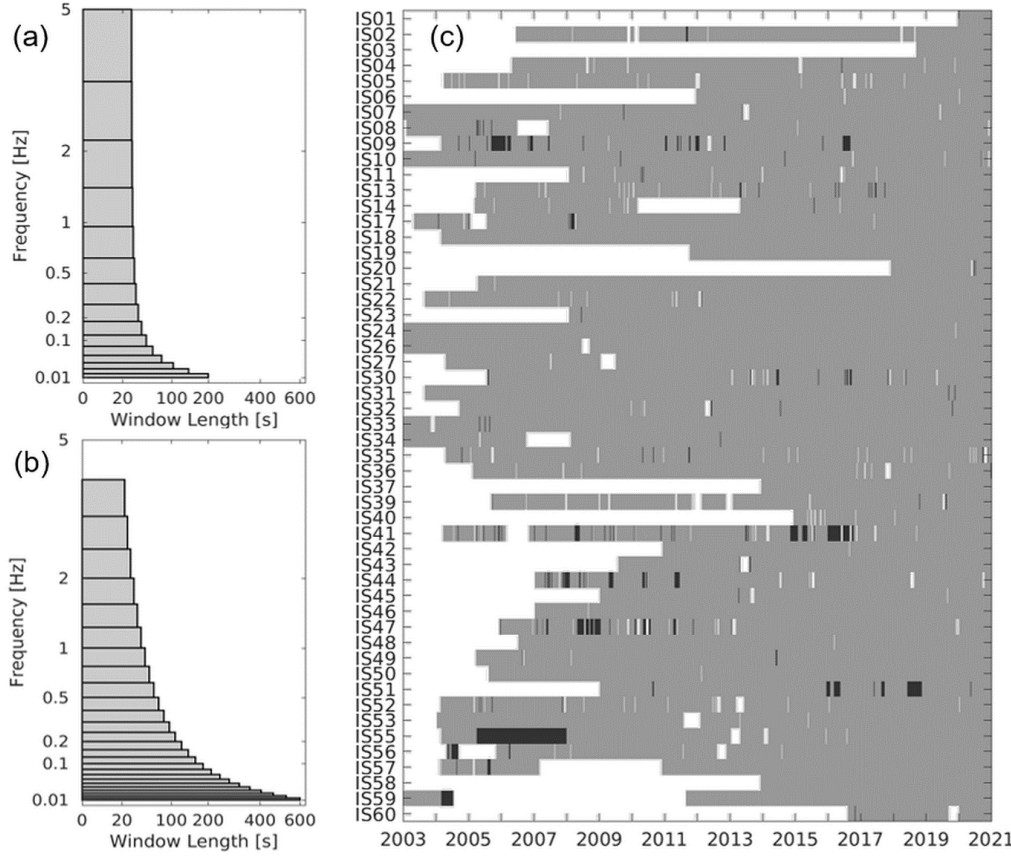

**Figure 2: Comparison of the reprocessing configurations to run the PMCC algorithm; (a) log-spaced frequency bands as previously used by Ceranna et al. (2019) and (b) one-third octave frequency bands applied in this study. The data availability (c) of reprocessing results (grey) is depicted on a daily basis. Black lines indicate that raw waveform data were available but could not be processed, most often because the number of sensors was too low (<3). IS25 is not depicted because of less than one month of data availability in the considered period (certified in December 2020). Note that panel (c) refers to the database available at the German NDC. As**

**the data have been subsequently obtained from the IDC, the data coverage agrees with that available through the vDEC; some exceptions might apply as data gaps could be backfilled at the IDC, where available.**

PMCC records a detected arrival within a frequency band and time window as a pixel. The estimated arrival parameters comprise, for instance, the back azimuth denoting the direction of origin, the frequency, the root-mean-square (RMS) amplitude or the apparent velocity. The latter reflects the velocity of the wave front in the x–y domain and is generally larger

than the actual phase velocity because of the wave front's inclination. Pixels adjacent to others in terms of time, back azimuth,





and apparent velocity are grouped into detection families if at least 10 pixels contribute; i.e., PMCC assumes the same event to be the origin. The chosen maximum tolerances are 120 s, 10° to 5°, and 10 % to 5 % of the apparent velocity, respectively. The two latter depend on the frequency bands, accounting for a potentially lower resolution of the parameters at very low frequencies being a result of the array response. Hereinafter a detection family is referred to as a detection. In the detection list
for each station, a detection's wave front parameters are averaged over all contributing pixels. PMCC neglects arrivals that are not associated with the dominant detection in a time–frequency domain (Cansi, 1995).

To our knowledge, standard thresholds for grouping pixels have not been specified. The chosen minimum of 10 can be justified by significance in order to consider as many arrivals as possible (e.g., Che et al., 2019). Concerning an upper threshold, we constrain the family sizes to 200 pixels during the processing. This can split detections originating from a specific event, with
a few exceptions when PMCC merges two families (i.e., >200 pixels). A higher threshold could summarize an event in only one family, which could be useful for certain applications. On the other hand, a higher threshold could result in mixing different sources (e.g., microbaroms) or in creating families with a long duration. Both could be at the expense of a parameter's detail when, for instance, the azimuth smoothly changes over time, either due to a moving source or due to propagation effects. The chosen upper limit of the number of pixels is therefore a compromise, in addition to the tolerances given above. Another
potential source of splitting effects is the length of the data sequences processed at once. However, since we apply the processing to daily data files extended by ±5 min, an impact on the analysis is deemed marginal here. Nevertheless, detections of similar wave front parameters that might have been split by PMCC could be merged to 'events' during a dedicated post-processing.

We realized the reprocessing of all data for the period 2003 to 2020 using PMCC version 5.7.4 (CEA, 2018). This version has
turned out to perform faster and it apparently operates more reliably than previous versions, making it even more efficient. With the chosen configuration, processing the daily infrasound data of one station takes between 3 and 6 min on average (operating system: CentOS 6.10, 32 GB RAM, 8 CPUs @3.4 GHz). The processing time varies with array size and the number of signal detections.

We post-process the detection lists to discard obvious artefacts. These include spurious signals lining up at the center
frequencies of the processing bands (i.e., the mean frequency equals the center frequency) or at certain back azimuths. The majority of these signals are single-frequency-band detections (i.e., the minimum and maximum frequencies of a detection equal those of one particular frequency band). Seemingly, PMCC is sometimes unable to resolve the wave front parameters correctly, resulting in a ringing artefact. The latter is also known from previous reprocessing configurations and the concerned detections feature low family sizes. Here, we sort out detections exactly equaling any center frequency of the processing bands
as well as detections of which the minimum and maximum frequencies differ by less than 6 mHz; this second criterion regards detections only covering the two lowest frequency bands. In the same context, we also clean the detection lists by discarding detections with family sizes <40; for frequencies of <0.06 Hz, detections with <50 pixels are discarded. Effectively raising the lower family size threshold ensures the global comparability of the stations' detection lists and the derived products, even if the ringing artefact affected not all stations to the same extent. Notwithstanding the above, the apparent velocities must range





between 300 and 500 m s$^{-1}$ to be accepted as an infrasonic signature (e.g., Lonzaga, 2015). The availability of raw waveform data and processing results per station are shown, on a daily basis, in Fig. 2c.

## 3 Broadband detection lists

For the 53 IMS stations certified before the end of 2020 (see Fig. 1), the detection lists comprise a total of 81,267,602 entries over 18 years. In Fig. 3, the frequency distributions of all detections relative to the back azimuth (Fig. 3a) and the apparent

velocity (Fig. 3b) as well as a 1D histogram (Fig. 3c) are shown.

### 3.1 Processing results

As stated by Le Pichon et al. (2010) and Ceranna et al. (2019), the detections can be roughly classified into three groups.

(i) Almost 40 % of all detections are signals with frequencies above 0.5 Hz. Many of these are of a transient nature with frequencies beyond 0.8 Hz. The sources include, for instance, volcanoes, surf, supersonic aircraft, explosions and industrial

activity. Their origin can generally be found within a few hundreds of kilometers from the station location since the transmission loss at these frequencies limits the propagation range – according to Sutherland and Bass (2004), the molecular attenuation coefficient for a 2 Hz wave is two magnitudes larger than for a 0.1 Hz wave. Therefore, the high-frequency azimuthal distribution is strongly linked to the station locations, as can be recognized by various peaks in Fig. 3a. At stations on islands or in coastal environments, surf is often the dominant source in the high-frequency infrasound range up to 20 Hz.

Garcés et al. (2006) reported that surf energy can be found down to frequencies of 0.4 Hz, thus overlapping with the microbarom range. The main energy associated with surf, however, has generally been found between 1 and 5 Hz as was demonstrated for IMS stations IS59 in Hawaii (Garcés et al., 2003), IS57 in California (Arrowsmith and Hedlin, 2005), and IS24 on Tahiti (Le Pichon et al., 2004). The 2D histogram shown in Fig. 3b indicates that the majority of high-frequency detections are tropospheric or stratospheric arrivals because it maximizes at apparent velocities between 330 and 365 m s$^{-1}$.

Following Lonzaga (2015), apparent velocities higher than ~380 m s$^{-1}$ correspond to reflection altitudes in the thermosphere (>90 km), where the attenuation coefficient increases with altitude (Sutherland and Bass, 2004). Below 0.8 Hz, quasi-continuous sources, especially the growing impact of microbarom signals, lead to a smoother azimuthal distribution.

(ii) Almost 55 % of all detections have frequencies between 0.1 and 0.5 Hz. The maximum of the frequency distribution (Fig. 3c) reflects the dominant frequency of microbaroms at between 0.2 and 0.3 Hz. Explosions, meteorites, and volcanic eruptions

are among the sources (Ceranna et al., 2019), but the microbaroms are the dominant signal type as every IMS infrasound station detects them in the course of the year (e.g., Landès et al., 2012; De Carlo et al., 2021). In Fig. 3a, both a pattern of easterly directions and a more dominant pattern of westerly directions with increased detection numbers reflect the characteristic annual cycle of microbarom detections, controlled by the stratospheric winds. An example of how the direction of the detected signals correlates with the middle atmosphere wind conditions is presented in Fig. 4, showing the processing

results of IS27 in Antarctica. IS27 is located near the Antarctic Circumpolar Current (ACC), being the major source of

microbaroms in the Southern Hemisphere throughout the entire year, whereas high frequency sources are rare in this region. The predominant arrival directions of microbaroms change from northwest in the austral winter to northeast in the summer.

**Figure 3: (a) Frequency–azimuth and (b) frequency–velocity distributions (left panels) of all detections at 53 stations from 2003 to 2020. The right panels of (a) and (b) show the 1D histograms for azimuth and velocity, (c) for the frequency. In the 2D histograms, the solid black lines depict the frequency bands of the processing with widths of one-third octave (see Fig. 2b). The number of detections refer to bins of approximately 1/10 of the frequency bandwidths and 0.5° (a) or 0.5 m s⁻¹ (b), respectively.**



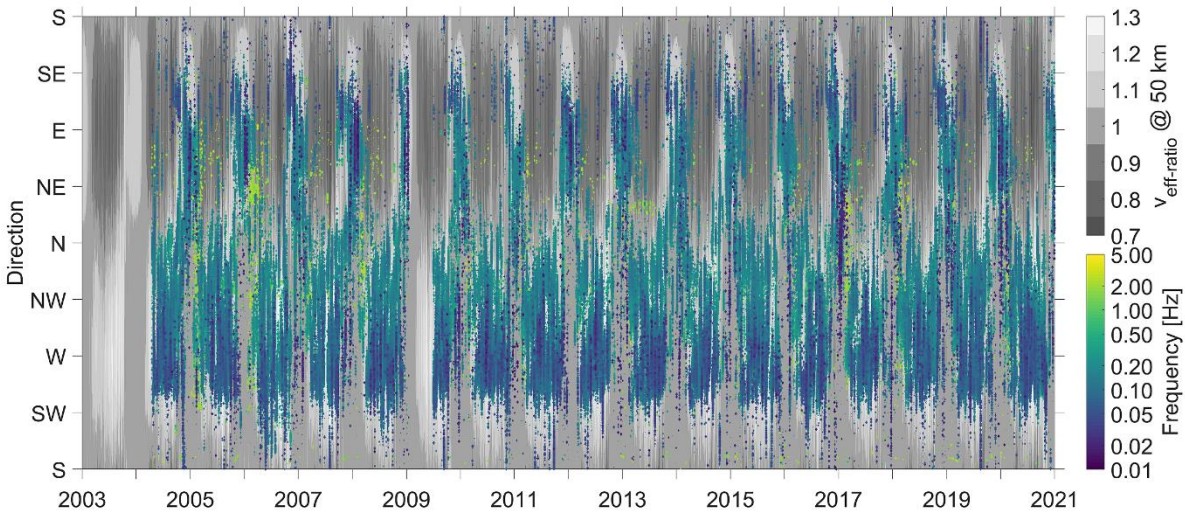

**Figure 4: Detections at IS27 from 2004 to 2020, with color-coded frequency. The lack of detections in 2009 is because of the array's relocation and revalidation (see Section 2.1). The $v_{\text{eff-ratio}}$ at around 50 km altitude (grey scale) was calculated from temperature and wind profiles obtained from ECMWF's HRES operational atmospheric model analysis as the maximum ratio between 40 and 60 km. Dark grey background colors ($v_{\text{eff-ratio}} < 0.95$) indicate unfavorable propagation from the respective direction, whereas favorable conditions in light grey ($v_{\text{eff-ratio}} \geq 0.95$) often coincide with an increased number of detections. The majority of detections are microbaroms (0.1-0.5 Hz) originating from the Antarctic Circumpolar Current (northwesterly to northeasterly directions).**

(iii) The minority (<6 %) of the detections are characterized by frequencies lower than 0.1 Hz. At these frequencies, infrasonic waves can travel long distances through the atmosphere. The corresponding phenomena, however, are either very rare – e.g., large meteorites like the Chelyabinsk fireball (Pilger et al., 2015) – or spatially confined to middle to high latitudes – e.g., MAWs or auroras. MAWs are the most frequent source of these phenomena and associated with strong tropospheric winds over mountainous regions (e.g., Hupe et al., 2019b); hence the station location relative to the sources becomes relevant again. Moreover, the transmission loss is reduced at these frequencies, limiting the importance of the stratospheric waveguide. Northerly and southerly directions (Fig. 3a) and higher apparent velocities (Fig. 3b) dominate the detections. It is of note that aurora infrasound is often characterized by apparent velocities above 500 m s$^{-1}$ (Wilson et al., 2010) and will therefore hardly be represented in the detection lists. Also, high apparent velocities can partly be caused by a reduced array response at low frequencies due to the geometry of the station, resulting in a poorer estimation of this parameter.

## 3.2 Comparison with the previously used PMCC configuration

We compare the frequency–azimuth histograms of the detection lists of the different configurations using the example of IS26, Germany. In Fig. 5a, all detections with family sizes >20 resulting from the 15-bands configuration (Matoza et al., 2013; Ceranna et al.; 2019) are processed for 12 years (2003–2014). It is noteworthy that the 15-bands configuration was limited to family sizes of 100 instead of 200 (both with exceptions), hence the lower threshold of 20. Ringing artefacts at the respective frequency band centers are discarded at the expense of some true detections, the latter because discriminating between both is not as straightforward as for the 26-bands configuration (Fig. 5b). With the newer configuration, we obtain around 82 % more

detections (in total 1.879 million) within the same time. Due to the narrower frequency bands, the average difference between the minimum and maximum frequency within a family has decreased from 1.09 to 0.82 Hz. This will have contributed to the increased number of detections. On the contrary, the average family size has increased from 72 to 169 pixels and the average

duration from 90 to 105 s, suggesting that the chosen one-third octave band configuration provides detections with a larger signal content. These enhanced features better highlight particular sources in the frequency–azimuth histograms, for instance a cluster at 0.3 to 0.6 Hz and 15° to 20° that can hardly be recognized in Fig. 5a. More detections enable an enhanced discrimination between sources, and as the processing artefacts can be subtracted more easily using the newer configuration, low-frequency sources are better represented. At IS26, two clusters around 0.7 Hz and southerly directions become more

evident. In the high frequency range, the newly implemented processing configuration neglects sources beyond 4 Hz (previously 5 Hz). This upper band threshold has been adapted to reduce the amount of local clutter and rather focus on signals that have propagated long range. Moreover, the accuracy (correlation) of the detections decreases with increasing frequency because of the corresponding wavelengths relative to the array apertures and the selected sub-networks (Section 2.2). A few sources might be lost due to this adaption, including small volcanic eruptions. Nevertheless, a maximum of around 4 Hz is

sufficient for capturing a global picture of the coherent ambient infrasonic noise as well as relevant events in the context of CTBT monitoring and verification. Overall, the example of IS26 shows that the newly implemented processing configuration with PMCC version 5.7.4 outperforms the formerly used one with PMCC 4.4.

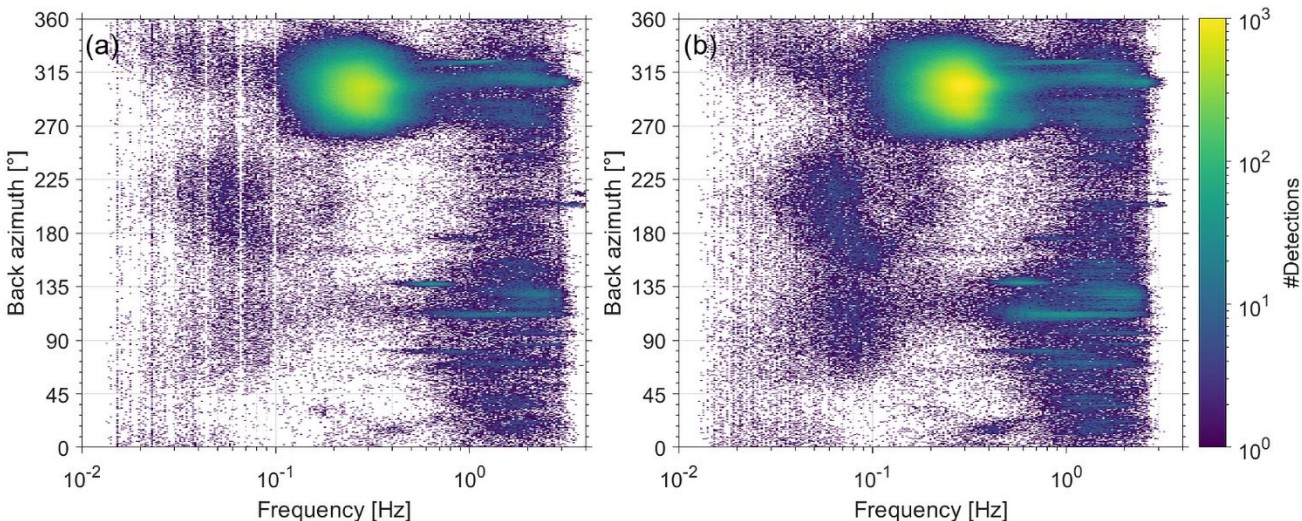

**Figure 5: 2D histograms of broadband detection lists for IS26, Germany, based on (a) 15 logarithmically spaced bands (Matoza et**
**al., 2013; Ceranna et al., 2019) and (b) 26 one-third octave bands for the PMCC configuration. The bin sizes equal those of Fig. 3 (1/10 of the one-third octave frequency bands and 0.5°). The detections of (a) were processed using PMCC version 4.4 and cover the years 2003–2014; (b) is built upon the more recent PMCC version 5.7.4 – the temporal coverage is adapted to that of (a) for comparability. The white vertical lines near the center frequencies in (a) result from cleaning the detection list of ringing artefacts; with the newer version and configuration, the cleaning is easier to narrow down to the respective center frequencies.**


### 3.3 Data quality parameter

As stated in Section 2, missing sensors or other array issues can affect the processing performance and thus the detection parameters. The number of available sensors, $N_{avail}$, as well as the number of sensors contributing to a PMCC detection, $N_{contr}$, are crucial indicators for the quality of a detection. When comparing long-term time series, changes in the array size, $N_{max}$, also have to be taken into account. For the quality of a detection, two PMCC output parameters are considered. The correlation coefficient, $r_c$, denotes the correlation of an arrival between the contributing sensors; here, $r_c$ is taken as the mean over all arrivals within a family. The Fisher ratio, $F$, puts the variance of noise plus the coherent signal in relation to the variance of noise; hence, $F$ exceeds one only if a coherent signal is recorded (Melton and Bailey, 1957). The Fisher correlation analysis in the time domain (e.g., Fisher, 1992) can be an alternative method to PMCC for detecting coherent signals within uncorrelated noise, but PMCC computes $F$ for each detection as an additional output parameter.

We combine all of these parameters empirically to define a single parameter as a proxy for the relative quality, $Q$:

$$Q = \frac{1}{2}\left(r_c \cdot w_{r_c}(f) + \frac{N_{contr}}{N_{max}}\right) \cdot \frac{F}{N_{avail}-1} \qquad (1)$$

which is the arithmetic mean of the two summands in the parentheses times the Fisher ratio relative to the available sensor number. For this calculation, we weight the correlation coefficient by a frequency-dependent function, $w_{r_c}(f)$. This function compensates the fact that low-frequency detections have a larger correlation coefficient due to the narrower frequency bands than high-frequency detections (Fig. 6).

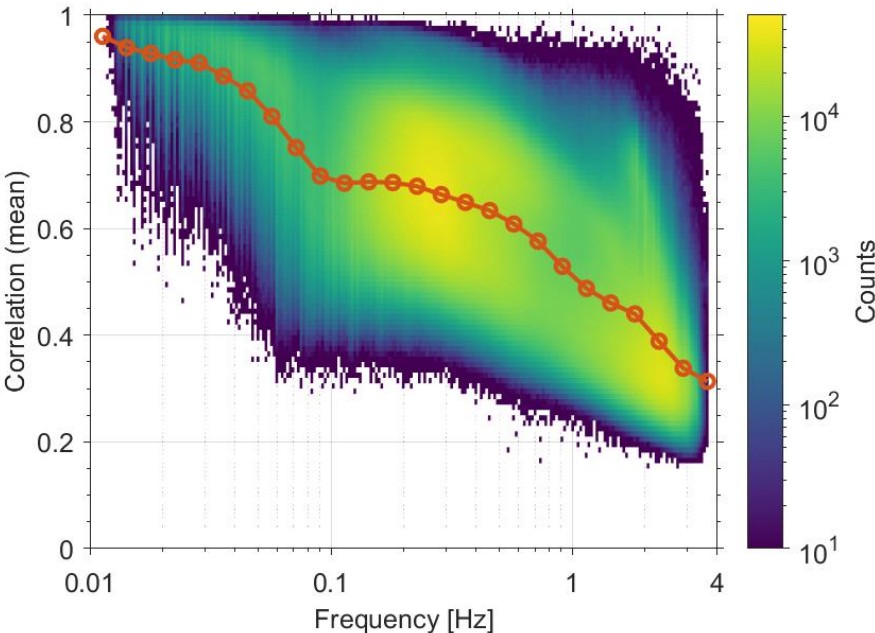

**Figure 6: Correlation coefficients of all detections (2003–2020, all stations) vs. frequency. The red line depicts the mean correlation for each of the 26 frequency bands. Bin sizes are one tenth of a frequency band and 0.01, respectively. The frequency band dependent weighting function $w_{r_c}$ in Eq. 1 weights the correlation coefficients of all detections such that the mean correlation of each frequency band would equal 0.5.**





$Q$ ranges between zero and one, as the correlation and the relative sensor availability (i.e., the first two summands of the parentheses in Eq. 1) do. In the case of a large $F$, the last factor of Eq. 1 can exceed one; if this results in $Q > 1$, we define $Q = 1$. Considering $Q$ enables assessing the quality of the detections – the lower $Q$, potentially the lower is at least one of the weighted correlation coefficient, the relative sensor coverage, or the Fisher ratio. We do not implement a threshold for $Q$ as

the cause for a reduction in any of these parameters can be manifold (e.g., GPS clock failure, flooded sensor, background noise level). Moreover, an anomaly can affect only specific detection parameters while others remain unaffected and may still be of use. Figure 7 shows such an example for IS05 in Australia. Here, it is noteworthy that the Fisher ratio also correlates with the frequency, similar to the correlation coefficient. Nevertheless, we do apply a weighting like for the correlation coefficient, as $F$ seems appropriate to indicate anomalies in certain parameters. At IS05, the back azimuths (Fig. 7a) follow the annual cycle

of the propagation conditions similar to Fig. 4, and these detections have generally been consistent since 2004. The amplitudes (Fig. 7b), however, exhibit anomalies until 2012, and in both 2018 and 2019 even more detections feature particularly large amplitudes (near 1 Pa) before returning to the normal range ($10^{-3}$ to $10^{-1}$ Pa). The anomalously large amplitudes correspond to relatively low values of $Q$ ($Q < 0.12$), which is here caused by very low Fisher ratios. Anomalies in the back azimuths or the frequencies are not recognized during that period. According to the IDC in Vienna, temporary issues of the WNRS for one of

the sensors were reported in early 2019, which potentially caused the observed anomalies here.

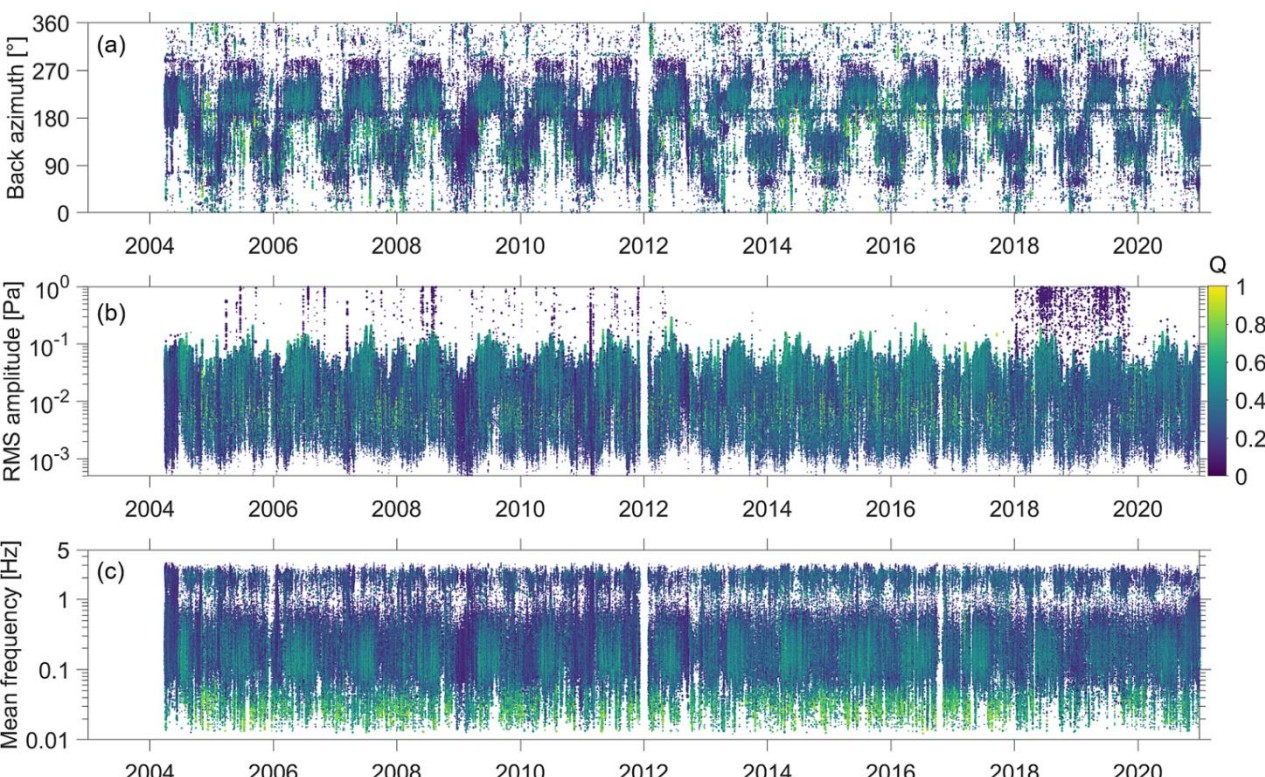

**Figure 7: Time series of the detection parameters back azimuth (a), amplitude (b), and frequency (c) for IS05, Tasmania (Australia). The color bar applies to all panels and depicts the quality parameter $Q$, defined in Eq. 1. Figure C1 in Appendix C shows IS05 detections with $Q>0.5$ only.**



In general, $Q$ tends to be a bit higher at lower frequencies (Fig. 7c) because $F$ is not weighted like $r_c$. Another feature in Fig. 7 are low values of $Q$ in late 2008 and early 2009 while lacking obvious anomalies in any of the shown parameters. The congruent drop to $Q < 0.4$ can be traced back to lower $F$ again, accompanied by the fact that fewer sensors, temporarily only 4 out of 8, contributed to the PMCC detections.

## 4 Data Products

From the comprehensive detection lists, we define four data products that cover different time–frequency domains. Here the time reflects the temporal resolution of the data set, as we provide detection parameters at distinct, equally spaced time steps. The time step interval matches the window length; hence there is no overlap. Note that we define the data products based on the mean frequencies of the detections. The minimum and maximum frequencies of the detections (families) are not a criterion and can thus cover the entire frequency range of the processing. The defined products are tailored to specific phenomena and

spectral peaks in the mean-frequency distribution (Fig. 3c).

(1) A low frequency product (Hupe et al., 2021a) incorporates detections with mean frequencies of between 0.02 and 0.07 Hz, which encompasses only 3.5 % of all detections. The time step and window length are 30 min. Since MAWs are likely the mostly detected source in this frequency range (e.g., Hupe et al., 2019b), we name this product '*maw*'.

(2) The second product covers the spectral peak near 0.25 Hz in the frequency distribution of all station detections (Fig. 3c).

The frequency range of this product is 0.15 to 0.35 Hz (36 % of all detections). Since the majority of the detections are (low-frequency) microbaroms, the product is named '*mb_lf*' (Hupe et al., 2021b). The temporal resolution is 15 min.

(3) This product ranges from 0.45 to 0.65 Hz (10.1 %), which includes the upper frequency spectrum of microbaroms and is therefore named '*mb_hf*' (Hupe et al., 2021c). The temporal resolution is 15 min.

(4) The secondary spectral peak around 2 Hz (Fig. 3c) is covered by the high-frequency ('*hf*') product (Hupe et al., 2021d)

with mean frequencies between 1 and 3 Hz and a temporal resolution of 5 min. One quarter (25.5 %) of all detections contribute to this product, where surf is likely the mostly detected source.

For the first product, all family sizes ≥50 are considered to ensure a consistent threshold over this product's frequency range after cleaning the detection lists from artefacts. For the other products, the threshold of 40 holds, which matches the threshold for cleaning the detection lists from artefacts at frequencies beyond 0.06 Hz (see Section 2.2). In total, these four products

cover almost 75 % of all detections.

### 4.1 Parameters

A parameter provided at a time step summarizes the parameters of 'dominant' detections within the time window since the previous time step. To identify 'dominant' detections, 1D back azimuth histograms are considered in which the family sizes of the detections are stacked per 1°. Stacking the family sizes by azimuth and over the time windows compensates a large

portion of the potential split of detections caused by the processing configuration threshold of 200 (Section 2.2), except from



the transition at the discrete time steps. The location of the maximum of the histogram defines the dominant direction (i.e., the most signal content arrived from that direction). If more than one maximum exists, the direction with the highest $Q$ scores off the others. All potential detections within ±5° tolerance from the dominant direction are then considered for calculating the mean back azimuth weighted by family size, which is consequently very close to the dominant direction. Based on the same

detections and weighting, further selected detection parameters (see Table 1) are accordingly summarized and represent the dominant detections. The majority of the parameters contain information about the weighted standard deviation of all detections within the time window respective to the calculated (dominant) mean. This standard deviation (detection-based) is detached from the one provided by PMCC (pixel-based), which is not considered here. For some parameters, deviating or additional quantities are provided, as detailed below.


**Table 1: Overview of the data product parameters provided in the netCDF files for each station. All 'mean', 'min', 'max' and 'sum' values correspond to all detections within the dominant azimuth ±5°. The 'mean' values are weighted by family size of the detections. The standard deviation ('std') accounts for all detections within the considered time window and refers to the calculated 'mean'. *N_time* denotes the total number of time steps, whereas *N_avail* denotes the number of time steps with product parameters available.**

| Parameter | Unit | Variable name | Type | Size | Description |
|---|---|---|---|---|---|
| Time (all time steps) | | time | char | N_time x 15 | [yyyymmddTHHMMSS (ISO 8601)] |
| Time (product available) | | time_p | char | N_avail x 15 | [yyyymmddTHHMMSS (ISO 8601)] (where num>0) |
| Duration | s | t_dur | double | N_avail x 1 | [sum] |
| Back azimuth | ° | azim | double | N_avail x 2 | [mean, std] |
| Apparent velocity | km s$^{-1}$ | vapp | double | N_avail x 2 | [mean, std] |
| RMS amplitude | Pa | a_rms | double | N_avail x 3 | [mean, std, max] |
| Frequency | Hz | freq | double | N_avail x 2 | [mean, std] |
| Family size | | f_size | double | N_avail x 3 | [mean, std, sum] |
| Correlation | | corr | double | N_avail x 3 | [mean, min, max] |
| Fisher ratio | | fish | double | N_avail x 3 | [mean, min, max] |
| Peak-to-peak amplitude | Pa | a_p2p | double | N_avail x 1 | [max] |
| Period at max. amplitude | s | p_max | double | N_avail x 3 | [mean, min, max] |
| Number of detections | | num | double | N_time x 2 | [at dominant azimuth ±5° \| within time window] |
| Quality parameter | | Q | double | N_avail x 1 | [max] - quality parameter accounting for sensor (data) availability, fisher ratio, and mean correlation |
| Sensor flag | | flag | double | N_time x 1 | flag for daily sensor availability: 1 - all sensors available, 2 - fewer sensors available, but at least three; 3 - less than three sensors, no PMCC detection possible |
| Sensor statistics | | sens | double | N_avail x 2 | [array size \| max. #sensors contributing to PMCC detections] |




### 4.1.1 Time parameters

All products contain a time vector of length *N_time*, which is the number of equally spaced time steps from 1 January 2003 to 1 January 2021. Another time vector ('*time_p*') is of length *N_avail*; it defines the time steps for which the product parameters are defined. The evaluated parameters refer to the time window since the last nominal time step (i.e., for the 'maw' product 405 with a resolution of 30 min, for instance, the time stamp of 1 January 2021 00 UTC summarizes all detections since 31 December 2020 23:30 UTC).

For the duration, neither the mean nor the standard deviation is calculated; instead, the durations of all detections with the dominant azimuth tolerance range are summed up during the time window. Implying that these detections originate from the same source, the total duration of the dominant signals is more conducive for the source characterization than the mean 410 duration. Moreover, this method partly compensates a potential bias if detections where split by the configured family size threshold.

### 4.1.2 Wave front parameters

Following the definition of the back azimuth being dominant in terms of family size, the tolerance range, and the weighting described in Section 4.1, the back azimuth parameter represents the dominant direction of a signal within the time window. 415 The standard deviation is calculated with regard to the determined mean. Both quantities have in common that they are weighted by the family size of the detections to account for the signal content and thus the significance of the single detection. As distinct to the mean, the standard deviation is calculated over all detections within the time window to illustrate the variability of the detections. Consequently, the standard deviation in back azimuth can exceed the tolerance range of ±5°.

The mean and the standard deviation are also provided for the frequency, the apparent velocity, the RMS amplitude, and the 420 family size. Here, the weighted mean values are again based on those detections that match the dominant azimuth ±5°; whereas the standard deviation with regard to the respective mean accounts for all detections within the time window weighted by family size. The family size variable additionally contains the sum, analogous to the duration, as this conveys an impression of how strong or long lasting the potential source was.

The RMS amplitude variable additionally contains the maximum amplitude of the dominant direction. The peak-to-peak 425 amplitude variable is exclusively provided as the maximum value within the dominant direction ±5°, as this will be of interest for yield estimations of (chemical) explosions. For instance, the Los Alamos National Laboratory (LANL) yield relation requires the wind-corrected amplitude and distance as input (e.g., Whitaker et al., 2003), and the maximum amplitude of the related detections likely results in a more realistic estimation than the mean.

The period at the maximum amplitude also serves as an input parameter to energy and yield estimations – for instance, two 430 relations of the Air Force Technical Application Center (AFTAC; ReVelle, 1997) are commonly used to estimate the energy of fireballs entering and burning up in the atmosphere (e.g., Ott et al., 2019; Pilger et al., 2020). The product variable of this

parameter contains the mean, the minimum, and the maximum of the dominant detections, enabling both lower and upper yield estimates and thus accounting for uncertainties.

### 4.1.3 Quality parameters and missing values

For the correlation and the Fisher ratio, also the mean, minimum, and maximum are provided. For $Q$, it is the maximum only. The additional variables '*flag*' and '*num*' are the only parameters that are defined at each time step even if the product is not defined, for instance due to unavailable raw data or the lack of processing results. The variable '*flag*' provides an indication of whether data of all sensors were available (*flag*=1). If at least three but not all sensors were available, it is *flag*=2. If fewer than the required three sensors were available, *flag* has the value 3. Then the variable '*num*' is zero. Otherwise, '*num*' indicates

the number of detections per time window (first column of '*num*') and with the dominant azimuth ±5° (second column), which are contributing to the standard deviations and means, respectively. The first column of the '*sens*' variable denotes the nominal array size (number of sensors), which remains constant or increases, but is not supposed to decrease; i.e., station revalidation periods are not accounted for. The second column is a number between 3 and the array size ($N_{max}$) denoting the maximum number of sensors contributing to the dominant PMCC detections. Overall, these variables enable to assess the quality of the

data products or to comprehend missing values.

### 4.1.4 Station parameters

Each product file contains the longitude, latitude, and elevation (1 x 1) of the respective station as variables '*lon*', '*lat*', and '*elev*'. The coordinates (WGS84) are rounded to the second decimal and allow users who are not familiar with the IMS network to apply the parameters, e.g. for triangulation, or to collect atmospheric model profiles of temperature and wind for the sites

and along the propagation path between the infrasound station and the anticipated or determined source location.

### 4.2 Format

All parameters are stored in netCDF files with the extension '.nc', including metadata about the parameter variables (Table 1). One file per infrasound station, year, and product is released. The file sizes depend on the number of time steps for which a product is available. They do not exceed 4 MB (*maw* product), 8.5 MB (*mb* products) and 25 MB (*hf* product), respectively.

The file names include the IMS infrasound station, the year, the abbreviated designation of the data product and its frequency range as well as the temporal resolution, representing the time step and window length (example: 'IS26_2013_mb_lf_0.15-0.35Hz_15min.nc'). For details on how to retrieve the data products, please see Section 6.

### 4.3 Temporal coverage

The maximum number of time steps per data product (*N_time* in Table 1) results from the considered period (2003 to 2020)

and the temporal resolution of the data products. Table B1 in Appendix B lists the portion of time steps at which a product is available per station (*N_avail/N_time*). Overall and at the majority of the stations, the *mb_lf* product is the one covered the



best, on average 23.5 %. Among due to the operational time of a station (see Figs. 1b and 2c), its detection capability, or its location relative to sources, this portion varies between 1.1 % (IS51, Bermuda) and 52.2 % (IS04, Western Australia). Such variability also applies to the other products, of which the overall coverage is lower than 10 %.

## 5 Case sensitivity and potential applications of the data products

For showing the capability of the data products, we choose both global and event-based approaches. We focus on two major explosive events that were of special interest for demonstrating the detection capability of the IMS infrasound network, as well as recent volcanic eruptions, being a significant natural hazard to civil security around the globe. For the coherent ambient noise, a global view of the data products is provided.

### 5.1 Warehouse explosion in the port of Beirut, Lebanon, on 4 August 2020

One recent event of interest is the devastating explosion in the port of Beirut on 4 August 2020, as the explosive yield was estimated in the order of the IMS network's design goal, using different waveform and remote sensing technologies (Pilger et al., 2021a). Infrasound detections were reported at five IMS arrays at distances of between 2,450 and 6,250 km from the source. The first signals arrived at IS48 in Tunisia and IS26 in Germany between 17:06 and 17:30 UTC. It is of note that the dominant frequencies differed at these stations, which is reflected in the mean values of 2.57 and 0.70 Hz, respectively. The frequency ranges of the multiple arrivals theoretically cover the *mb* and *hf* products, but the mean frequencies of the associated detection families do not. As the products are based on the latter, IS26 detections of the Beirut explosion are represented in the two *mb* products (17:15 and 17:30 UTC), whereas IS48 detections can be found in the *mb_hf* (17:30 UTC) and *hf* (17:10, 17:15, 17:20, 17:25, 17:30 UTC) products (Fig. 8a). The maximum peak-to-peak amplitudes provided in the products correspond approximately those values found by Pilger et al. (2021a), hence enabling reasonable yield estimates using the LANL yield relation (Whitaker et al., 2003) when additionally incorporating the atmospheric wind conditions.

### 5.2 Fireball near Chelyabinsk, Russia, on 15 February 2013

To date, the large meteorite and its fragmentation at around 30 km altitude, which according to Brown et al. (2013) released an explosive energy equivalent of roughly 500 kt TNT equivalent, produced the strongest signal that has been recorded by the IMS infrasound network, with 20 out of 42 operational stations detecting it (Le Pichon et al., 2013). At the closest station (540 km), IS31 in Kazakhstan, signatures of the large fireball arrived around 30 min after the impact at 03:20 UTC and covered a broad frequency range. As illustrated in Fig. 8b, the event is represented in four data products of IS31 (*maw, mb_lf, mb_hf, hf*). The first signatures from 25° slightly deviated in back azimuth from the theoretical one (23.7° from IS31 to 54.85°N, 61.45°E, i.e. south of Chelyabinsk; see Pilger et al., 2015). The back azimuth deviation increased with time, which allows inferring the trajectory of the fireball (e.g., Pilger et al., 2020). Combining the information of the different data products would



also enable to assess the temporal variation in azimuth in this case, encouraging the investigation of the propagation and source characteristics.

Apart from the IS31 detections, at other detecting stations long-period infrasonic waves were the dominant ones excited by the fragmentation and travelled very long distances. At IS53 in Alaska, for instance, the signals were still recorded after circling

the globe twice (Le Pichon et al., 2013). At stations more distant than IS31, such as IS43 in Russia (1,510 km) or IS26 in Germany (3,280 km), the explosion-like event was detected with frequencies below 0.1 Hz. The *maw* product includes these signatures, e.g. for IS26 at 07:30 UTC from 54.8° (standard deviation 1.7°), with the total duration of five detections being 585 s and $Q = 0.94$. For IS43, two entries in the *maw* product at 05:30 and 06:00 UTC sum up to a duration of 41 min (2,461 s) distributed over 19 detections. The minimum and maximum values for the period at the maximum amplitude are in the order

of those reported by Le Pichon et al. (2013). For IS26, for instance, our product gives 18.6 s and 30 s (Le Pichon et al.: 20 s and 35 s), respectively, and the mean is 24.8 s. Translated into yield estimates using the AFTAC relation (ReVelle, 1997), a TNT-equivalent range of between 90 kt and 640 kt results, while the mean corresponds to 290 kt. This range gives an impression of the uncertainty of the empirical yield relations, both at one station and when considering a set of stations. Le Pichon et al. (2013) eventually used the mean values of 14 stations for specifying the yield of the fireball to roughly 460 kt

TNT.

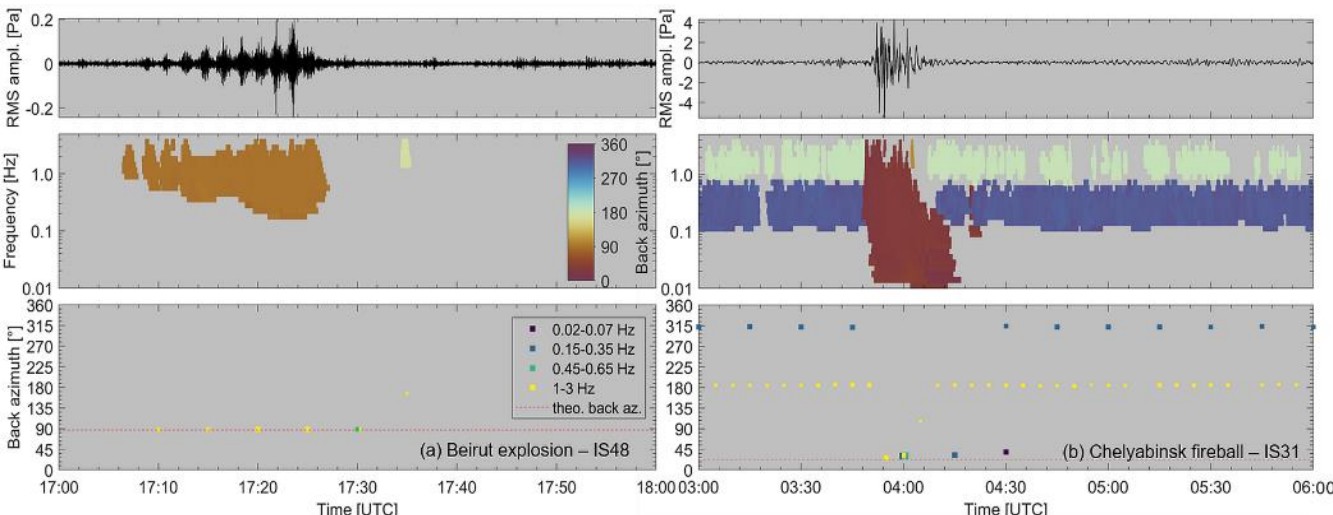

**Figure 8: Waveform beam (top), PMCC result (center), and data products for (a) the Beirut explosion detected at IS48, Tunisia (2,450 km distance), on 4 August 2020 and (b) the Chelyabinsk fireball detected at IS31, Kazakhstan (540 km), on 15 February 2013. The waveform beams correspond to the theoretical back azimuths and 3rd order Chebyshev bandpass filters (0.5-4 Hz and 0.02-2**

**Hz, respectively). The legends of (a) apply to (b), too. The relative marker size of the products (bottom panels) depicts Q. The dashed red lines depict the theoretical back azimuths from an array to the source. It is of note that the *mb_lf* and *hf* signatures caused by the fireball interrupt the series of IS31 detections from a northwesterly direction (~315°), most likely microbaroms, and from a southerly direction.**

Overall, infrasound recordings can be helpful to detect, locate and characterize large meteorites entering uninhabited regions

where they remain unnoticed by the public, such as the Bering Sea fireball on 18 December 2018 (Redd, 2019), which was the





second strongest fireball ever detected by the IMS infrasound network (e.g., Pilger et al., 2020). Infrasound observations can serve as a primary source of information or complement observations by other technologies (e.g., Ott et al., 2019).

### 5.3 Volcanic eruptions

One of the recent volcanic eruptions was Taal in the Philippines, beginning on 12 January 2020 and lasting for 10 days (Global Volcanism Program, 2013). It was assigned a Volcanic Explosivity Index (VEI) of 4 and thus comparable with the Eyjafjallajökull eruption on Iceland in 2010, which heavily affected global air traffic due to its ash plume. The explosive eruption phases of the Eyjafjallajökull volcano in April and May were recorded by 4 infrasound arrays of the IMS (IS18, IS26, IS43, IS48) at distances of between 2,200 and 3,700 km, plus 10 regional infrasound arrays in Europe within 2,200 km from the source (Matoza et al., 2011b). As an example, we remark that especially the *mb_hf* and *hf* products of IS26 in Germany contain a series of corresponding detections between 18 April and 02 May 2010.

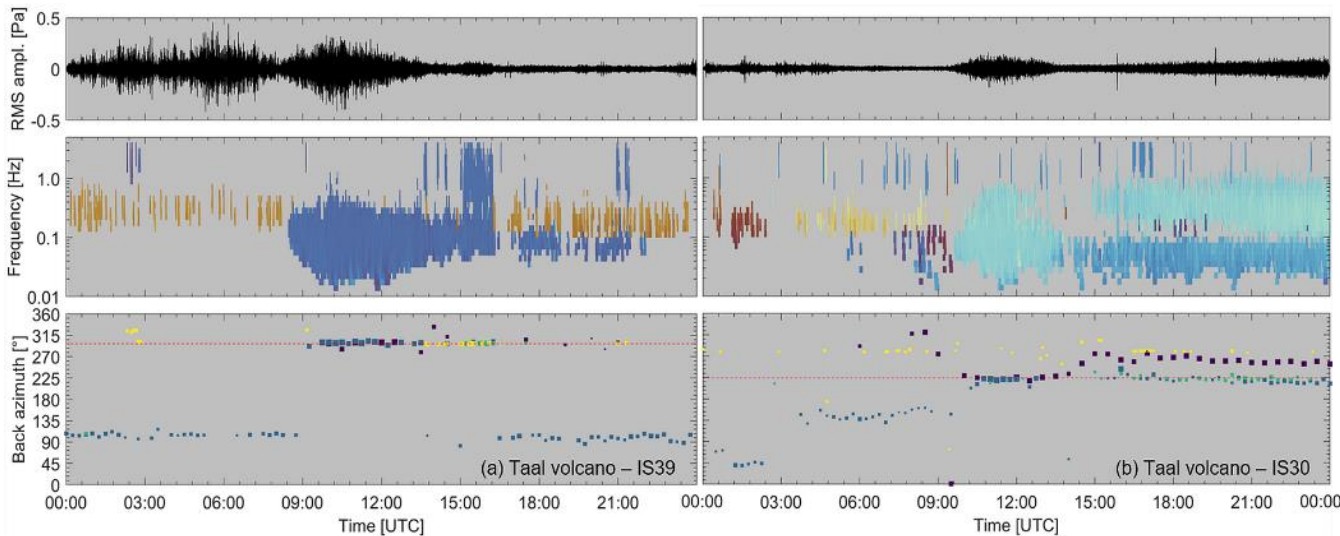

**Figure 9: Waveform beam (top), PMCC result (center), and data products of (a) IS39, Palau (1,640 km distance), and (b) IS30, Japan (3,050 km), on 12 January 2020, when a strong eruption of the Taal volcano (Philippines) occurred. The waveform beams correspond to the theoretical back azimuths and a 3rd order Chebyshev bandpass filter (0.05-4 Hz). The dashed red lines depict the theoretical back azimuths relative to Taal volcano. At IS39, the broadband signatures cover all data products. The colors and symbols of Fig. 8 apply.**

For the Taal volcano eruption in 2020, Perttu et al. (2020) found infrasonic signatures at IS39, Palau, which they used to estimate the plume height to 17 km maximum. Such information is essential for running ash dispersion models and thus forecasting the ash plume in the atmosphere, allowing both air traffic control and Volcanic Ash Advisory Centres (VAACs) as well as local authorities to implement timely safety measures (e.g., Matoza et al., 2019, and references therein). Figure 9a shows the back azimuths of the IS39 data products on 12 January 2020. The back-azimuths of arrivals beginning around 09 UTC are consistent with the direction of Taal (297°). The onset is distinguished in the *mb_lf* product because the initial direction of dominant detections suddenly changes by about 180°, followed by *hf* detections until 16:30 UTC. In addition to

IS39, we have found consistent detections at IS30, Japan. The IS30 data products (Fig. 9b) reproduce the detections well,
particularly the *maw* and *mb_lf* products. After 15 UTC, the *mb* products of IS30 show features within 224±15° that possibly
indicate ongoing activity beyond the end of that day.

Several other active volcanoes are covered by our data products, including the explosive eruptions of Calbuco, Chile (see
Matoza et al., 2018), Stromboli, Italy (Le Pichon et al., 2021), and Raikoke, Kuril Islands (McKee et al., 2021), to name a few
examples that were recently reported. Besides singular events, the IMS also captures quasi-continuous eruptive activity, among
which are the serial explosions of two volcanoes – Lopevi and Yasur – in Vanuatu (e.g., Le Pichon et al., 2005). These are
represented in the *hf* products of IS22, New Caledonia. Figure 10 shows the back azimuths of the IS22 products in 2020, with
*hf* signatures from 45° almost throughout the whole year being associated with the Yasur volcano at around 400 km distance.
Slight deviations from the theoretical directions are caused by crosswinds; therefore these signatures remain useful for probing
the atmospheric propagation conditions (Le Pichon et al., 2005), now over a long period and subsequently approaching
climatological time scales.

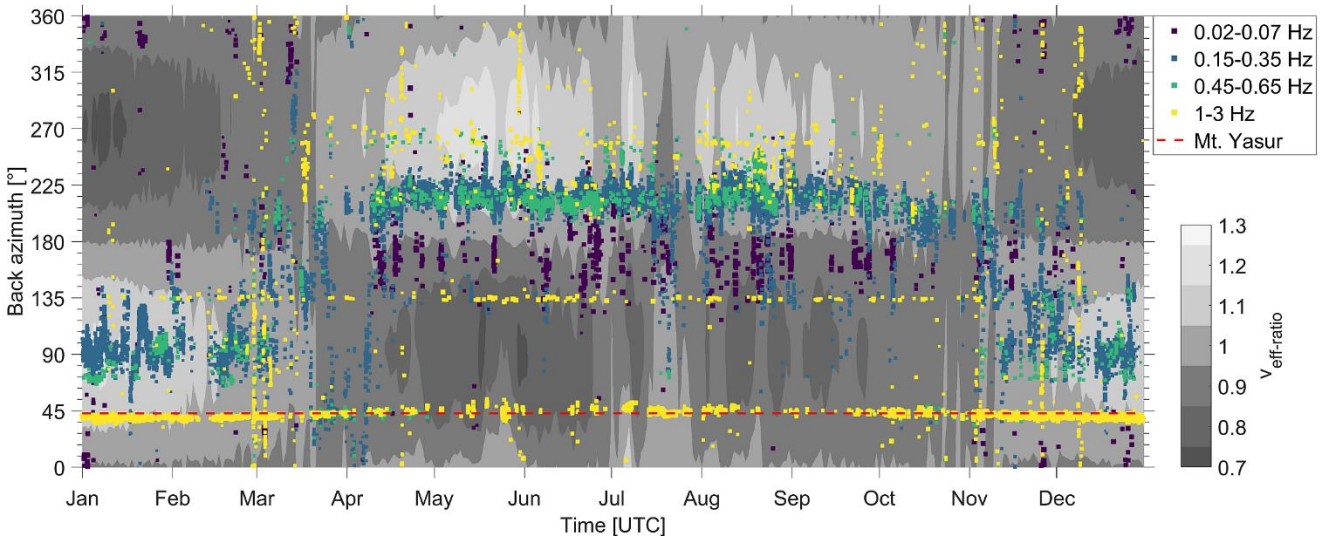

**Figure 10: Back azimuth of IS22 data products (color-coded) in 2020, with the $v_{eff-ratio}$ (grey scale, analogous to Fig. 4) shown in the background. If the $v_{eff-ratio}$ is around or larger than one, arrivals from the corresponding direction are favorable in terms of the local propagation conditions.**

Overall, we assume that our open-access data products of up to 18 years for 53 IMS infrasound stations can be utilized
systematically to monitor and characterize active volcanism around the globe, following the example set by Matoza et al.
(2017), who also incorporated the Eyjafjallajökull's explosive eruptions. Moreover, the data products can serve as an input to
and reference data set for recent advancements in early warning applications. For instance, the calculation of a volcano
infrasound parameter (IP) at regional and local arrays around Mt. Etna (Ripepe et al., 2018) has successfully been applied to
more distant IMS arrays by Marchetti et al. (2019), demonstrating that the IP enables warnings prior to explosive eruptions
that can be useful to VAACs. As the IP is a product of the mean infrasonic amplitude and the number of detections in a certain



time window, our data products provide the relevant information for its calculation. Since these products are not available in real time at this stage, a next step will be to demonstrate the capability of integrating the data products in such early warning applications based on past eruptions. This will pioneer the early release of tailored data products in the future.

## 5.4 Coherent ambient noise and atmospheric dynamics

The detections of microbaroms are related to the annual cycle of the middle atmosphere winds controlling the effective sound speed, and thus the directivity of the ground-to-stratosphere waveguide (e.g., Landès et al., 2012). De Carlo et al. (2021) recently applied the comprehensive detection lists described in Section 3 for validating an updated microbarom source and propagation model that was developed by De Carlo et al. (2020). Our objective is that the data products defined in Section 4 can also serve as such a reference database for future model developments. The temporal resolution of the $mb$ data products (15 min) is higher than that of commonly used atmospheric models (1 h or more) or the ocean wave model on which the De Carlo et al. (2020) model relies (3 h).

To demonstrate the products' capability, we reproduce the global microbarom activity using the $mb\_lf$ product of the 52 stations with at least one year of data availability – thus excluding IS25 – in Fig. 11. The roughly weekly variability of the mean dominant back azimuths (color-coded) is superimposed on the $v_{\text{eff-ratio}}$ that illustrates the stratospheric waveguide conditions for zonal propagation (back azimuth 270°), with light grey colors depicting favorable conditions for eastward propagation. It can be assumed that dark grey background colors represent favorable westward propagation conditions. The global overview highlights the annual reversal of the stratospheric winds as the direction of detections generally follows this pattern. It resembles the global picture given by Ceranna et al. (2019) who provided such illustration for the entire frequency range covered by the processing. This underlines that the microbarom detections constitute by far the mostly detected phenomenon within the IMS infrasound network, even if other events as those described in the previous subsections are not subtracted from the $mb\_lf$ product here. For a comparison, in Appendix C we provide Fig. C2, which qualitatively hardly differs from Fig. 11 at first sight, but relies on the comprehensive detection lists of Section 3 and covers a broader microbarom frequency range (0.1-0.5 Hz). The chosen time step (4 d) and window length (8 d) are identical.

In the Southern Hemisphere, the dominant arrival direction switches between the southwest (austral winter) and southeast (austral summer), as the ACC produces major microbarom sources throughout the year (e.g., De Carlo et al., 2021). Therefore the detectability mainly depends on the stratospheric waveguide direction. In the Northern Hemisphere, a clear pattern only establishes during the winter, while arrival directions relate to the northern parts of the Atlantic and Pacific Oceans, corresponding to southwesterly back azimuths at high-latitude arrays and northwesterly back azimuths at almost all other stations. In the summer, the westward stratospheric jet is weaker than the eastward jet during the winter, and the distribution of the continental landmasses additionally prevents a clear pattern.

When showing the global view based on the $mb\_hf$ product (Fig. C3 in Appendix C), the overall picture is similar, but some station-specific differences can be recognized. Most strikingly, IS30 in Japan exhibits a seasonal pattern in the $mb\_hf$ product that is not recognizable in the other figures, which can be explained by its location relative to the microbarom sources. The





Pacific Ocean to easterly directions is obviously dominant for the long-period microbaroms throughout the year, as it is nearly contiguous with the station. At shorter periods, smaller and marginal seas can be a relevant source of microbaroms where low-frequency microbaroms lack. At IS30, the Sea of Japan or the East China Sea are potential candidates for the arrivals from westerly directions in the winter. Such an effect likely causes differences at other stations, too. Therefore, the data products enable to discriminate between such sources, which may be useful for microbarom model validations or, for instance, studies

focusing on marginal seas. Moreover, these *mb* products can be of use for assessing atmospheric models, as microbarom detections can reveal uncertainties in the representation of the middle atmospheric temperature and winds (e.g., Hupe et al., 2019a), as well as temporary changes of the middle atmosphere conditions such as during SSW events (e.g., Assink et al., 2014).

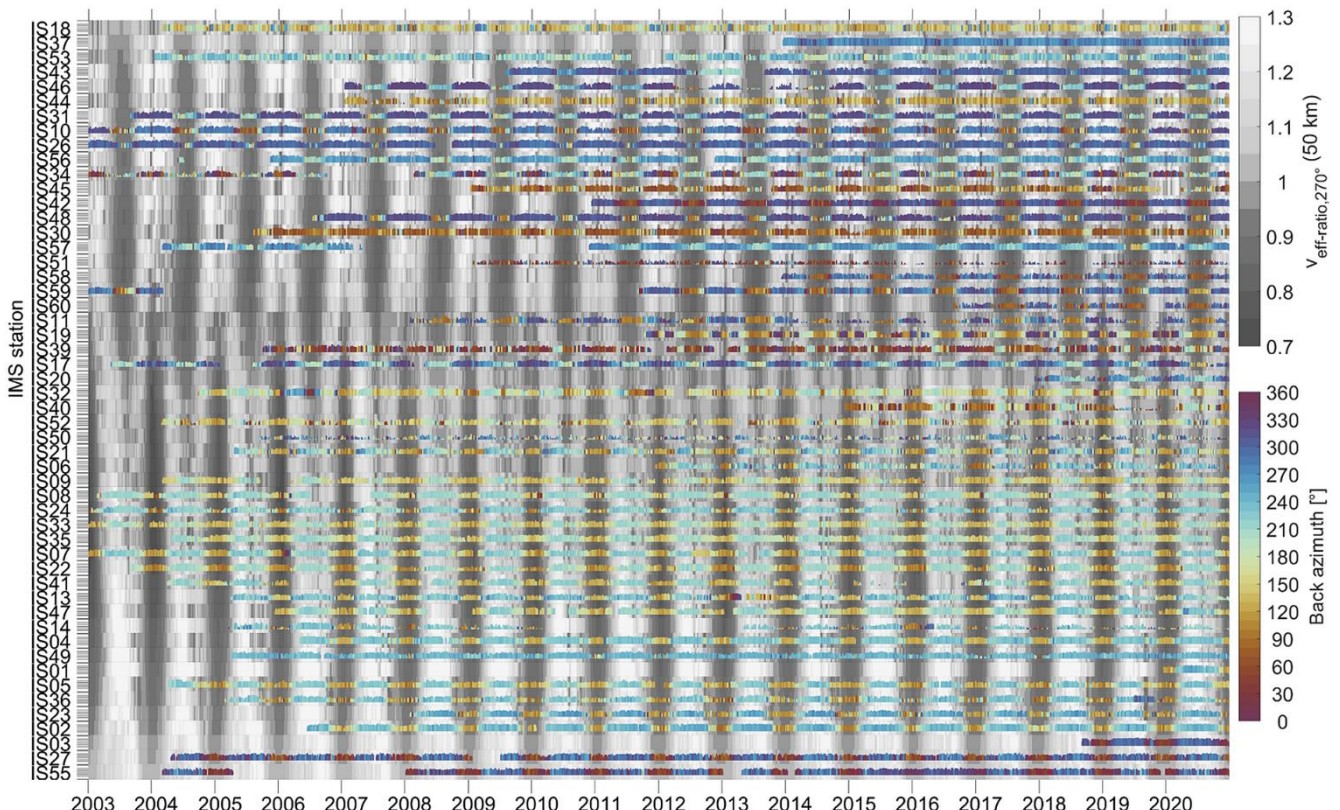

**Figure 11: Global picture of the back azimuth variation in the *mb_lf* product (0.15-0.35 Hz), averaged in time windows of 8 d. For each time step (4 d), the mean back azimuth is color-coded. The grey scale depicts the $v_{\text{eff-ratio}}$ calculated from the ECMWF temperature and wind profiles at each station analogously to Fig. 4, but for arrivals from westerly directions (270°). IS20 marks the transition from the Northern to the Southern Hemisphere.**

Analogously to Fig. 11, we illustrate the *hf* product in Fig. 12. At the majority of the stations, the *hf* product is also related to

the annual cycle of the stratospheric wind directions, depicted by the $v_{\text{eff-ratio}}$, such that the dominant directions often change with the wind reversals during the equinoxes. However, the dominant directions significantly differ from those of the *mb* products. As stated in Section 3, signals in the higher frequency range more likely originate from regional and local sources





because the propagation range is more constrained than for very low frequency acoustic waves. The difference is particularly striking at stations located in the Southern Hemisphere, where many IMS arrays are installed on islands or close to the coastlines and therefore surf is a major source. For instance, the dominant back-azimuth at IS24, Tahiti, is associated with westerly to southwesterly directions and hardly changes with the stratospheric wind reversal, which agrees with the observations by Le Pichon et al. (2004). Other island stations such as IS06, IS13, IS14, and IS23 also exhibit a relatively consistent dominant direction over the year, indicative of sources in the near field. In the Northern Hemisphere, this applies to the coastal station IS19 in Djibouti and IS59 in Hawaii. The latter is consistent with the northwesterly directions observed by Garcés et al. (2003), who found a correlation between the infrasonic amplitude and the wave height. Arrowsmith and Hedlin (2005) confirmed this correlation for IS57 in California. They noted the difference that the observed signals propagated over a longer range of around 200 km and they determined a seasonal cycle in the number of surf detections from northwesterly directions, with a maximum in the winter season. This agrees with the dominant directions shown in Fig. 12. In the summer season, signals from southeasterly directions prevail, corresponding to the stratospheric circulation. Besides surf, a frequently occurring source can be industrial noise. For instance, the dominant back azimuth (around 280°) at IS30 in Japan corresponds to the (southern) Tokyo Bay with the industrial area of Yokohama and Kawasaki and several power stations in the near field.

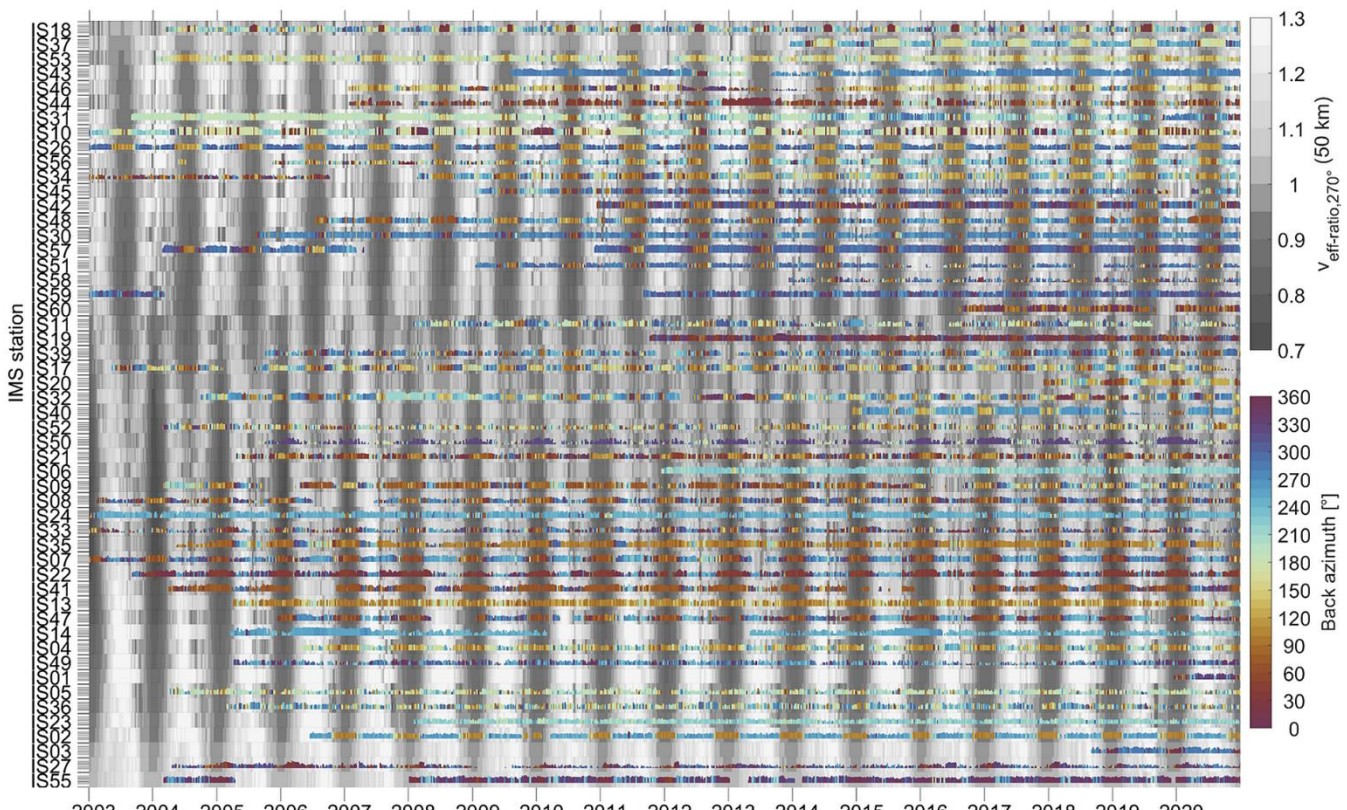

**Figure 12: Similar to Fig. 11, but for the *hf* product (1-3 Hz).**

Finally, the *maw* product is shown in Fig. 13, also following the examples of the global microbarom detections. It qualitatively
represents the global view provided by Hupe et al. (2019b); here we can use a higher resolution since the updated processing
results in a larger number of detections at low frequencies (Section 3). Hupe et al. (2019b) already noted that a seasonal pattern
can also be recognized for the MAW detections but the location of the arrays relative to the global MAW source regions is
more relevant and evident in the back azimuths. For a comparison, Fig. C4 in Appendix C reproduces Fig. 13 based on the
comprehensive detection lists. The main difference is the number of detections or non-NaN values, which results from the
larger time window of the product summarizing the dominant detections. The *maw* product may be of use for revisiting the
source localization and further investigating source excitation of MAWs, following the comprehensive study of MAW
detections at IMS infrasound detections by Hupe (2018), which was based on PMCC version 4.4.

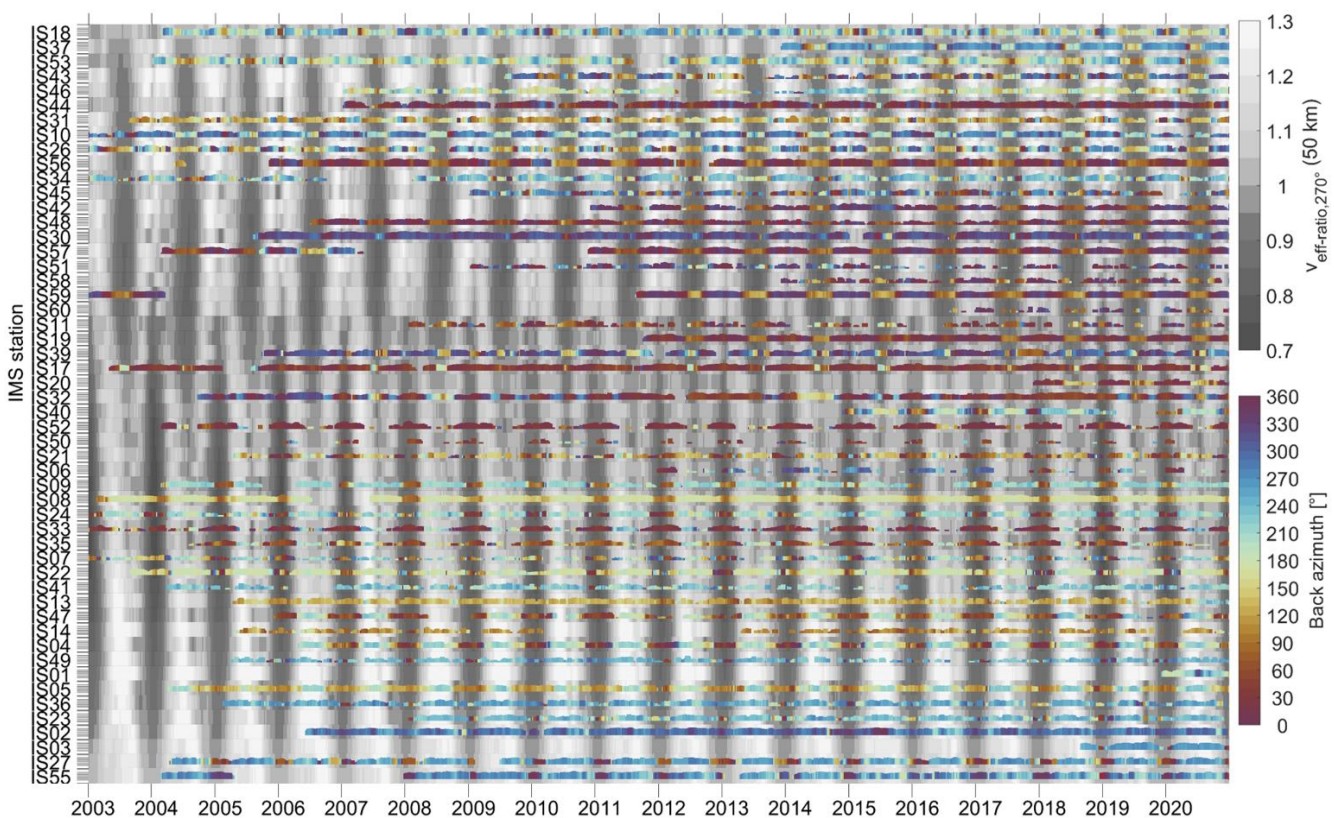

**Figure 13: Global picture of the back azimuth variation in the *maw* product (0.02-0.07 Hz), averaged in time windows of 30 d. For**
**each time step (7 d), the mean back azimuth is color-coded. The grey scale depicts the $v_{eff\text{-}ratio}$ calculated from the temperature and**
**wind profiles (ECMWF) at each station, for arrivals from westerly directions (270°). IS20 marks the transition from the Northern**
**to the Southern Hemisphere. Note that the vertical extent of each colored line, which denotes the number of non-NaN entries at a**
**logarithmic scale, is not directly comparable to that of Fig. 11 (different scaling).**



## 6 Data availability

The infrasound data products are openly accessible through the product center (https://produktcenter.bgr.de, last accessed on 8 December 2021) of the Federal Institute for Geosciences and Natural Resources (BGR), the German NDC (Pilger et al., 2017). For each of the four product types, the product center contains one so-called data series that is assigned a DOI. Each data series includes, on a yearly basis, a product entry with the respective data products for all stations compressed within one .zip file. The .zip files contain a README file as well as simple MATLAB code that reads and plots the netCDF data.

BGR's product center (in the future: 'Geoportal') ensures the long-term availability of the data products, while the Creative Commons Attribution 4.0 International license (CC BY 4.0, https://creativecommons.org/licenses/by/4.0/, last accessed on 8 December 2021) applies. Therefore, any reference related to the use of the data products should cite this accompanying manuscript and the respective DOI of the product type, stating the year and station (or summarize this information appropriately), e.g. for the *hf* products 'Hupe et al. (2021d), DOI:10.25928/bgrseis_bbhf-ifsd, stations IS26 and IS27, 2011–

2020'.

The DOIs of the different data series are as follows. The list of associated data sets is displayed when clicking the icon "show datasets".

- *maw* product series:  https://www.doi.org/10.25928/bgrseis_bblf-ifsd (Hupe et al., 2021a)
- *mb_lf* product series:  https://www.doi.org/10.25928/bgrseis_mblf-ifsd (Hupe et al., 2021b)
- *mb_hf* product series:  https://www.doi.org/10.25928/bgrseis_mbhf-ifsd (Hupe et al., 2021c)
- *hf* product series:  https://www.doi.org/10.25928/bgrseis_bbhf-ifsd (Hupe et al., 2021d)

Access to the IMS network's data including raw waveform recordings of the infrasound stations can be granted upon request through the virtual Data Exploitation Center (vDEC) of the IDC at https://www.ctbto.org/specials/vdec (last accessed on 29 October 2021).

Information about the high-resolution (HRES) operational atmospheric model analysis of the European Centre for Medium-Range Weather Forecasts (ECMWF), produced by the Integrated Forecast System (IFS), can be found on https://www.ecmwf.int/en/forecasts/datasets (last accessed on 29 October 2021). The relevant data used to calculate the $v_{eff\text{-}ratio}$ are available in ECMWF's Archive Catalogue (https://apps.ecmwf.int/archive-catalogue/, last accessed on 8 December 2021), which is published under the CC BY 4.0 licence.

## 7 Conclusions

Gaining a global picture of the coherent infrasound wave field is essential in terms of the CTBT verification and beneficial for probing the dynamics in the middle atmosphere where operational observation methods are sparse. The updated processing configuration with one-third octave frequency bands and the extended period of consideration advance the global reference data set of infrasound detections. The derived data products summarizing the comprehensive, highly resolved detection lists

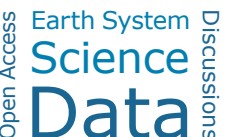

are created to make infrasound data accessible for a broader community. Four products of different temporal resolutions and frequency ranges can serve as reference database for various applications. These encompass model developments or validations within the infrasound and atmospheric community, infrasound data assimilation for atmospheric models, or early warning systems for civilian security.

The definition of the products signifies that only the dominant detections within a time window are represented by the open

access database. Moreover, other methods than the PMCC algorithm may be more appropriate to obtain a comprehensive picture of the azimuthal soundscape because PMCC itself focuses on the dominant signal within a time window and frequency band. For instance, CLEAN beamforming (e.g., Den Ouden et al., 2020) and the vespagram method (Vorobeva et al., 2021) have been demonstrated to be capable of capturing multiple sources at selected IMS stations. Nevertheless, PMCC is a well established processing method in the infrasound community. The comprehensive detection lists described in this study have

already been used for the validation of a microbarom model (De Carlo et al., 2021). The dominant infrasound signals will likely represent those signatures that are relevant for civilian security applications, too. Examples of recent volcanic eruptions show that the different data products cover these and other transient events appropriately.

The openly accessible data products of detection lists do not replace the privileges of either a vDEC access to raw infrasound data or nominated users who have access to the Reviewed Event Bulletin of the IDC. The tailored data products, in terms of

frequency ranges and temporal resolution, instead simplify the utilization of infrasound data by user groups who are not experienced in the use of array processing methods such as the PMCC algorithm. Additional parameters provided with the products enable to assess the relative quality of the data. We therefore assume that this archived database of currently 18 years can contribute to advancing the use of infrasound observations for scientific applications and the development of prototypes. The introduced quality parameter will be further elaborated on, while in parallel work considering station and time dependent

confidence intervals on wave parameters is ongoing. These so-called PMCC metrics will be published separately to help identifying anomalies in the data and its products. It is envisaged to update the products' database regularly, which means at least an offset of three months with respect to the current vDEC regulations that allow to access raw data after this embargo time. For early warning systems (e.g., volcanic eruptions) or atmospheric model assimilations, however, it would be an asset to provide the tailored data products near real time in the future.




**Appendices**

**Appendix A: IMS infrasound station information**

**Table A1: IMS infrasound array properties and selected sub-networks for the PMCC processing. If elements were added to the array, the year of this modification (revalidation) and the previous number of array elements is indicated in parentheses behind the latest number of elements. These changes did not alter the apertures because the new sensors were installed within the existing arrays. The sub-network information contains triplets of sensors numbers, referring to the latest channel numbering (e.g., L1, … L4, … or H1, … H8, …). If two types ('H' and 'L') are still defined (IS18 was homogenized to 'H' in 2016, but the naming was not changed), the corresponding letter is indicated.**

| Array | lat [°N] | lon [°E] | elevation [m] | # elements | aperture [km] | Sub-networks |
|-------|----------|----------|---------------|------------|---------------|--------------|
| IS01 | -41.11 | -70.72 | 1013 | 8 | 1.51 | 2,4,5; 2,5,7; 5,6,8; 1,6,8 |
| IS02 | -54.58 | -67.31 | 94 | 5 | 1.43 | 1,4,5; 1,2,5; 1,2,3; 1,3,4 |
| IS03 | -68.58 | 78.07 | 64 | 7 | 1.40 | 3,5,6; 2,6,7; 1,3,4; 3,5,7 |
| IS04 | -34.60 | 116.36 | 167 | 8 | 2.41 | 1,2,3; 1,3,7; 1,2,7; 2,3,7 |
| IS05 | -42.49 | 147.68 | 350 | 8 | 2.24 | 1,3,5; 2,5,8; 2,4,7; 2,5,6 |
| IS06 | -12.15 | 96.82 | -10 | 8 | 1.56 | 1,3,5; 2,5,8; 2,4,7; 2,5,6 |
| IS07 | -19.93 | 134.33 | 386 | 8 | 2.26 | 1,2,3; 1,2,4; 1,3,4; 2,3,4 |
| IS08 | -16.22 | -68.45 | 4096 | 4 | 2.30 | 1,2,3; 1,2,4; 1,3,4; 2,3,4 |
| IS09 | -15.64 | -48.02 | 1185 | 4 | 2.26 | 1,2,3; 1,2,4; 1,3,4; 2,3,4 |
| IS10 | 50.20 | -96.03 | 247 | 4 | 2.60 | 1,2,3; 1,2,4; 1,3,4; 2,3,4 |
| IS11 | 15.26 | -23.18 | 44 | 8 | 1.97 | 2,6,7; 3,4,8; 2,5,6; 2,4,5; 1,7,8 |
| IS13 | -27.13 | -109.36 | 163 | 8 | 2.82 | 1,2,4; 1,2,3; 1,3,4; 2,3,4 |
| IS14 | -33.65 | -78.80 | 388 | 8 | 2.64 | 1,2,4; 1,3,4; 1,2,8; 2,4,6 |
| IS17 | 6.67 | -4.86 | 156 | 4 | 2.94 | 1,2,3; 1,2,4; 1,3,4; 2,3,4 |
| IS18 | 77.48 | -69.29 | 69 | 8 | 1.16 | 1,2,3; 1,2,4; 1,3,4; 2,3,4 (L) |
| IS19 | 11.47 | 43.17 | 21 | 8 | 2.01 | 1,2,3; 1,2,4; 1,3,4; 2,3,4 |
| IS20 | -0.60 | -90.37 | 448 | 8 | 2.06 | 4,5,6; 6,7,8; 4,5,8; 1,6,7 |
| IS21 | -8.87 | -140.16 | 806 | 4 | 1.83 | 1,2,3; 1,2,4; 1,3,4; 2,3,4 |
| IS22 | -22.18 | 166.85 | 272 | 4 | 2.15 | 1,3,4; 1,2,4; 1,2,3; 2,3,4 |
| IS23 | -49.35 | 70.24 | 84 | 15 | 1.60 | 4,7,10; 4,10,13; 1,7,10; 4,7,13 |
| IS24 | -17.75 | -149.30 | 117 | 5 | 3.89 | 1,2,3; 1,2,5; 1,4,5; 2,3,4 |
| IS25 | 16.13 | -61.63 | 245 | 9 | 1.85 | 2,4,5; 2,6,9; 1,7,8; 5,6,7 |
| IS26 | 48.85 | 13.71 | 1110 | 8 (2014: 5) | 2.57 | 1,2,3; 1,4,5; 1,3,4; 2,4,5 |
| IS27 | -70.70 | -8.30 | 50 | 9 | 1.92 | 3,7,8; 4,5,6; 1,6,7; 6,7,8 |
| IS30 | 35.31 | 140.31 | 52 | 6 | 2.12 | 1,5,6; 1,2,6; 1,2,4; 2,3,4 |
| IS31 | 50.41 | 58.03 | 367 | 9 (2019: 8) | 2.19 | 1,8,9; 1,7,8; 1,7,9; 7,8,9 |
| IS32 | -1.24 | 36.83 | 1662 | 7 | 1.13 | 2,3,4; 3,4,6; 2,4,6; 2,3,5 |
| IS33 | -19.01 | 47.31 | 1385 | 4 | 2.48 | 1,2,3; 1,2,4; 1,3,4; 2,3,4 |
| IS34 | 47.80 | 106.41 | 1245 | 8 (2008: 4) | 2.97 | 1,2,3; 1,3,4; 1,2,4; 2,3,4 |
| IS35 | -19.19 | 17.58 | 1253 | 7 | 2.05 | 3,4,7; 2,3,5; 1,2,4; 2,3,4 |
| IS36 | -43.92 | -176.48 | 14 | 8 | 2.86 | 5,6,7; 3,7,8; 5,7,8; 2,6,8 |
| IS37 | 69.07 | 18.61 | 74 | 10 | 1.95 | 3,4,6; 2,3,7; 1,2,4; 2,3,4 |
| IS39 | 7.54 | 134.55 | 100 | 7 | 2.44 | 3,4,6; 2,3,7; 1,2,4; 2,3,4 |
| IS40 | -4.30 | 152.01 | 85 | 8 | 1.87 | 1,3,4; 1,6,7; 1,7,8; 1,6,8 |
| IS41 | -26.34 | -57.31 | 164 | 4 | 2.28 | 1,2,3; 1,2,4; 1,3,4; 2,3,4 |
| IS42 | 39.04 | -28.01 | 284 | 8 | 1.51 | 1,2,3; 1,2,4; 1,3,4; 2,3,4 |
| IS43 | 56.72 | 37.22 | 120 | 6 | 1.55 | 1,2,4; 1,2,3; 1,3,6; 2,3,5 |





| | | | | | | |
|---|---|---|---|---|---|---|
| IS44 | 53.11 | 157.71 | 380 | 4 | 1.90 | 1,2,4; 1,3,4; 1,2,3; 2,3,4 |
| IS45 | 44.20 | 131.98 | 150 | 4 | 2.17 | 1,2,4; 1,3,4; 1,2,3; 2,3,4 |
| IS46 | 53.95 | 84.82 | 232 | 4 | 2.84 | 1,2,4; 1,3,4; 1,2,3; 2,3,4 |
| IS47 | -28.62 | 25.24 | 1306 | 8 | 1.69 | 2,4,8; 1,6,7; 3,5,6; 3,4,5; 2,7,8 |
| IS48 | 35.81 | 9.32 | 850 | 7 | 1.86 | 1,6,7; 4,5,6; 2,5,7; 5,6,7 |
| IS49 | -37.09 | -12.33 | 81 | 5 | 2.14 | 1,4,5; 1,2,3; 1,3,4; 2,3,4 |
| IS50 | -7.94 | -14.38 | 186 | 8 | 2.79 | 5,6,7; 5,6,8; 5,7,8; 6,7,8 |
| IS51 | 32.36 | -64.70 | -30 | 4 | 2.40 | 1,2,3; 1,2,4; 1,3,4; 2,3,4 |
| IS52 | -7.38 | 72.48 | 1 | 7 | 1.90 | 3,4,5; 1,2,3; 2,4,7; 2,3,4 |
| IS53 | 64.88 | -147.86 | 200 | 8 | 1.97 | 3,4,6; 4,5,7; 1,2,7; 2,3,5 |
| IS55 | -77.72 | 167.65 | 45 | 8 | 2.83 | 4,5,7; 1,5,6; 3,4,8; 1,2,8; 2,3,6 |
| IS56 | 48.26 | -117.13 | 763 | 4 | 2.24 | 1,2,3; 1,2,4; 1,3,4; 2,3,4 |
| IS57 | 33.61 | -116.45 | 1248 | 8 | 1.45 | 1,2,3; 1,2,4; 1,3,4; 2,3,4 |
| IS58 | 28.21 | -177.38 | 5 | 4 | 1.86 | 1,2,3; 1,2,4; 1,3,4; 2,3,4 |
| IS59 | 19.59 | -155.89 | 1034 | 4 | 1.89 | 1,2,3; 1,2,4; 1,3,4; 2,3,4 |
| IS60 | 19.29 | 166.61 | 5 | 8 | 3.37 | 2,4,5; 2,3,8; 1,2,3; 3,4,7 |


## Appendix B: Data product availability

**Table B1: Product availability per station ($N\_avail$) in % of the total number of time steps over 18 years ($N\_time$, in parentheses).**

| Station / Product: | *maw* (315,600) | *mb_lf* (631,200) | *mb_hf* (631,200) | *hf* (1,893,600) |
|---|---|---|---|---|
| IS01 | 0.8 | 2.7 | 1.4 | 0.5 |
| IS02 | 12.1 | 48.8 | 14.1 | 8.5 |
| IS03 | 1.3 | 7.1 | 1.9 | 1.1 |
| IS04 | 4.8 | 52.2 | 14.6 | 6.4 |
| IS05 | 6.8 | 28.5 | 5.8 | 3.4 |
| IS06 | 0.4 | 8.6 | 12.2 | 14.4 |
| IS07 | 0.8 | 22.0 | 29.5 | 18.4 |
| IS08 | 10.4 | 27.9 | 3.5 | 5.5 |
| IS09 | 3.4 | 22.6 | 5.7 | 10.3 |
| IS10 | 4.8 | 34.1 | 6.6 | 34.5 |
| IS11 | 1.0 | 8.7 | 4.1 | 5.4 |
| IS13 | 2.7 | 17.5 | 4.1 | 19.6 |
| IS14 | 0.6 | 4.1 | 3.3 | 9.0 |
| IS17 | 9.0 | 20.3 | 2.4 | 10.4 |
| IS18 | 17.2 | 49.6 | 22.0 | 10.8 |
| IS19 | 5.4 | 15.0 | 10.6 | 6.9 |
| IS20 | 0.4 | 2.4 | 1.2 | 3.9 |
| IS21 | 2.1 | 25.8 | 4.1 | 6.8 |
| IS22 | 6.5 | 31.2 | 6.0 | 12.1 |
| IS23 | 1.6 | 15.2 | 1.5 | 3.2 |
| IS24 | 3.1 | 19.3 | 3.0 | 15.6 |
| IS25 | 0.1 | 0.2 | 0.1 | <0.1 |
| IS26 | 5.0 | 48.7 | 10.8 | 9.1 |
| IS27 | 6.9 | 33.3 | 7.4 | 1.4 |
| IS30 | 16.9 | 35.8 | 18.8 | 9.5 |
| IS31 | 3.6 | 25.7 | 3.4 | 24.0 |





| | | | | |
|---|---|---|---|---|
| IS32 | 9.0 | 49.7 | 37.2 | 16.7 |
| IS33 | 1.7 | 23.9 | 2.8 | 3.2 |
| IS34 | 5.8 | 15.3 | 5.4 | 13.8 |
| IS35 | 1.5 | 32.5 | 22.1 | 18.7 |
| IS36 | 2.3 | 15.3 | 9.8 | 8.1 |
| IS37 | 8.6 | 32.2 | 19.1 | 13.6 |
| IS39 | 7.0 | 26.7 | 8.4 | 6.9 |
| IS40 | 2.2 | 13.7 | 7.4 | 12.9 |
| IS41 | 1.0 | 14.6 | 7.5 | 14.3 |
| IS42 | 1.9 | 23.9 | 30.5 | 22.9 |
| IS43 | 2.6 | 23.8 | 5.1 | 15.1 |
| IS44 | 10.4 | 34.1 | 17.0 | 7.1 |
| IS45 | 1.9 | 15.2 | 3.2 | 4.2 |
| IS46 | 3.4 | 23.7 | 4.4 | 8.7 |
| IS47 | 3.2 | 35.5 | 17.9 | 6.8 |
| IS48 | 1.9 | 22.1 | 5.8 | 8.5 |
| IS49 | 0.8 | 14.2 | 2.0 | 2.1 |
| IS50 | 0.1 | 1.8 | 0.5 | 3.4 |
| IS51 | 0.4 | 1.1 | 0.3 | 1.1 |
| IS52 | 3.5 | 24.7 | 7.8 | 5.2 |
| IS53 | 22.6 | 41.7 | 16.1 | 12.3 |
| IS55 | 12.7 | 30.8 | 10.5 | 10.5 |
| IS56 | 20.5 | 37.7 | 6.3 | 8.0 |
| IS57 | 5.6 | 27.8 | 25.8 | 30.5 |
| IS58 | 0.4 | 5.2 | 1.5 | 0.4 |
| IS59 | 8.1 | 25.0 | 4.2 | 7.7 |
| IS60 | 0.1 | 3.3 | 1.7 | 5.8 |
| **overall** | **5.0** | **23.5** | **9.1** | **9.9** |



## Appendix C: Additional Figures

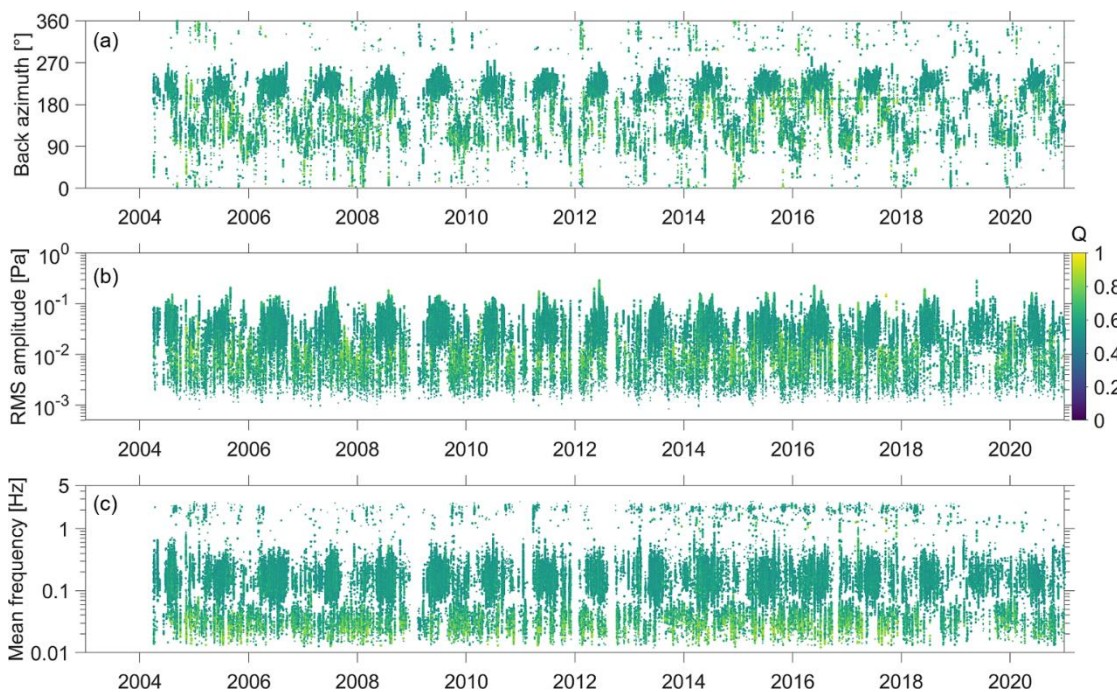

**Figure C1: As Fig. 7, but only incorporating detections with Q>0.5.**

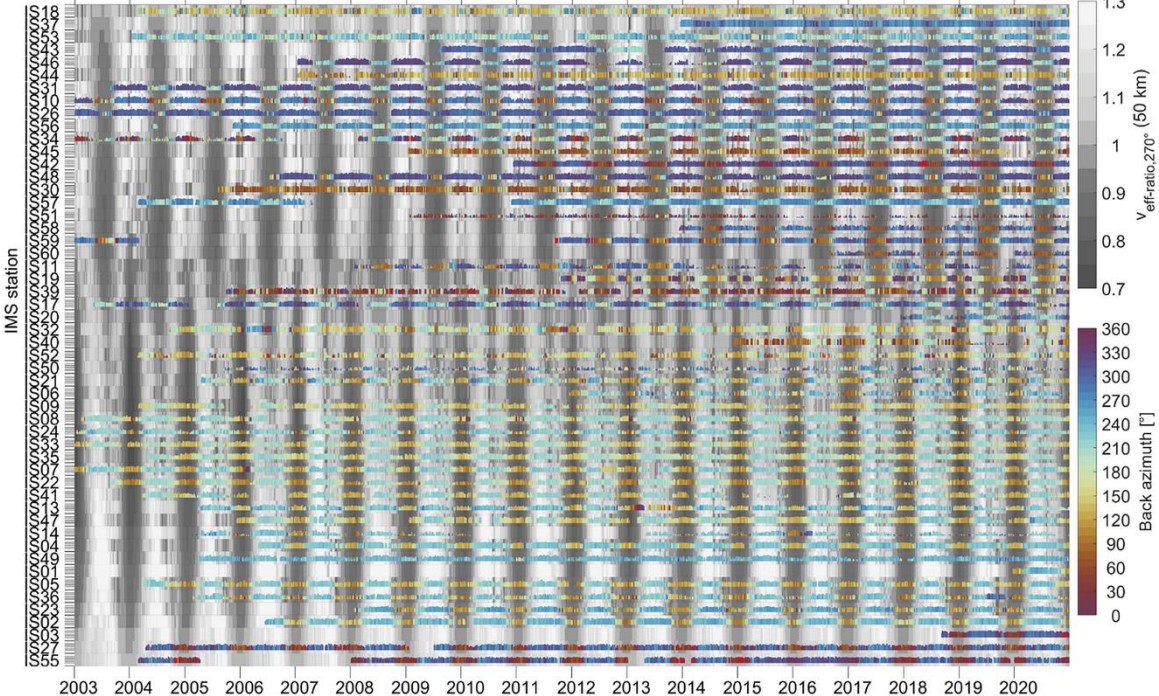

**Figure C2: Similar to Fig. 11, but for the IMS infrasound detections in the microbarom frequency range (here 0.1-0.5 Hz).**





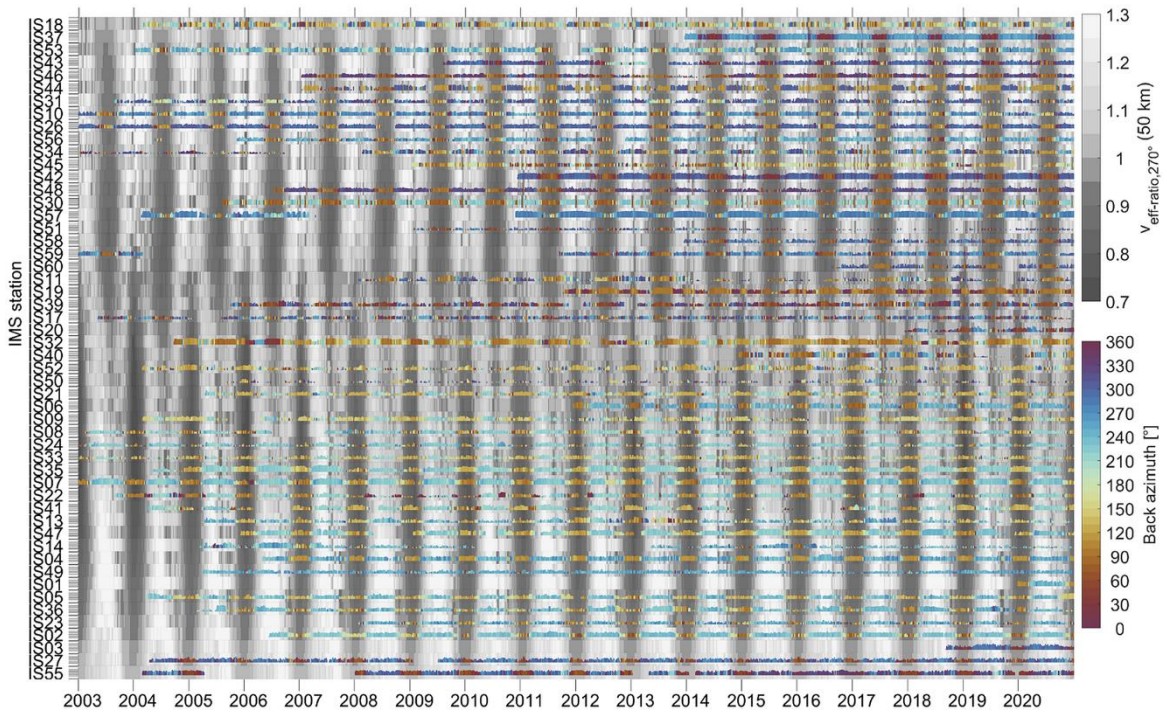

**Figure C3: As Fig. 11, but for the *mb_hf* product (0.45-0.65 Hz). As the vertical extent of each colored line denotes the number of non-NaN entries at a logarithmic scale, this global view is based on much less data than Fig. 11.**

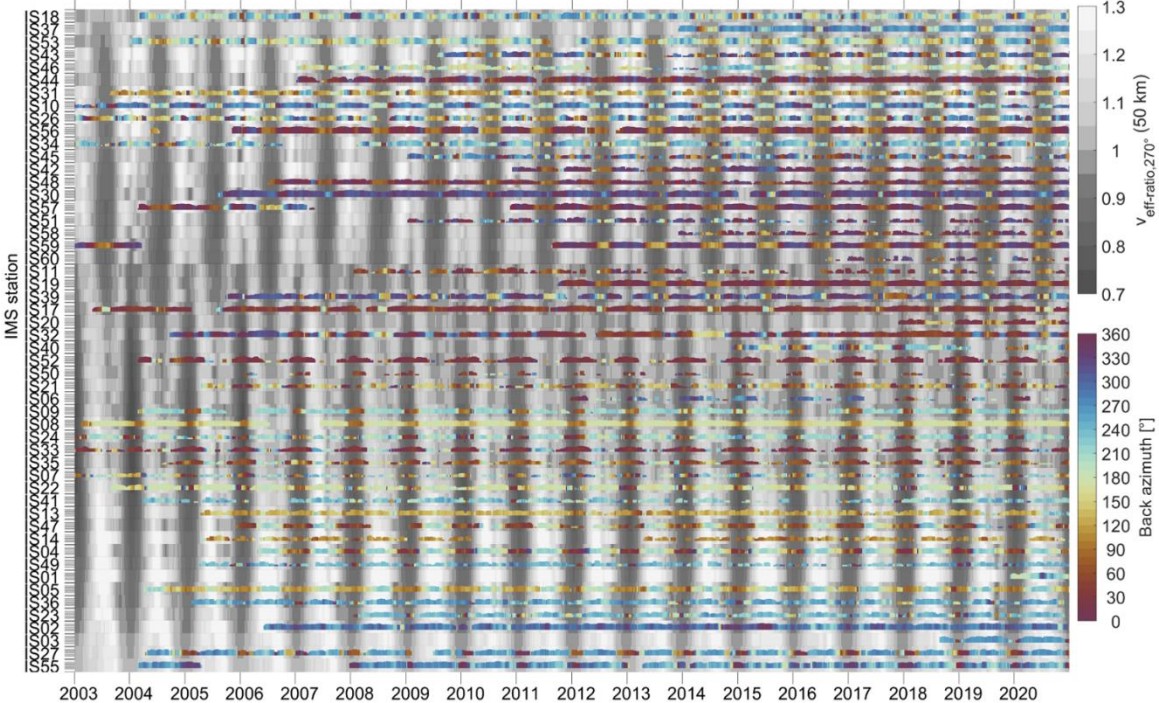

**Figure C4: Similar to Fig. 13, but for the IMS infrasound detections in the MAW frequency range (0.02-0.07 Hz).**



**Author contributions**

Conceptualization: PH, LC; Data curation: PH; Formal analysis: PH; Investigation: PH, LC, ALP, RSM, PM; Methodology:
PH, LC, ALP, RSM; Visualization: PH, LC, ALP, RSM; Writing – original draft preparation: PH; Writing – review & editing:
LC, ALP, RSM, PM.

**Competing interests**

The authors declare that they have no conflict of interest.

**Disclaimer**

The views expressed herein are those of the authors and not necessarily reflect the views of the CTBTO Preparatory
Commission.

**Acknowledgements**

All authors thank the CTBTO and the IMS station operators for guaranteeing the high quality of the infrasound data. PH
acknowledges the CTBTO Preparatory Commission for providing limited access (via vDEC) to the IMS infrasound network
data, which enabled to undertake this study. RSM acknowledges support from NSF grant EAR-1847736. The idea for the
infrasound data products initially aimed at a follow-up upon phase two of the ARISE project (Atmospheric dynamics Research
InfraStructure in Europe; phase 1: 2012–2015, phase 2: 2015–2018) but was pursued and matured independently. The
colormaps used within this manuscript are (i) 'romaO' (Crameri, 2018) where color scales depict back azimuths and (ii)
'viridis' (Biguri, 2020) for all other colored plots.

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
