# Peer review of "International Monitoring System infrasound data products for atmospheric studies and civilian applications"

_Earth System Science Data, 2021_

## Author Comment (AC1)

**Author reply to RC1 (essd-2021-441)**

Dear Catherine de Groot-Hedlin,

we appreciate that you took the time to carefully read and review our manuscript. Thank you very much for your positive recommendation. We reply to your specific comments below. We also provide guidance for accessing the downloadable data sets through produktcenter.bgr.de.

Yours sincerely,

Patrick Hupe on behalf of all co-authors
* * *
Comment: Data availability. I tried clicking on the produktcenter.bgr.de link but could not gain
access. Is the site experiencing broad technical difficulties or does it block certain areas of the world?

Reply: We cannot find out if or why the link https://produktcenter.bgr.de was inaccessible. We verified again that the link works on all common internet browsers (e.g., Firefox, Chrome, Edge).

Are the data available through that link or the doi.org links listed on about line 660? I could not find the icon "show datasets" as described on line 656. Will these become available when the paper is published?

The products are already available! There are two options to find the data products:

1) Through the landing page https://produktcenter.bgr.de, where you can search for the products. By typing, e.g., "infrasound products", a list appears that shows all related data series (e.g., "Higher frequency data products of the International Monitoring System's infrasound stations") and data sets (e.g., "hf_2003"). The product search can be refined by typing "*maw* product", "*mb_lf* product", "*mb_hf* product", or "*hf* product" for more specific lists of the respective products.
The data sets provide the actual download links ["Access" -> "Further information (Data)"], whereas the data series describe these data sets and metadata.
2) Through the doi.org links provided in the paper, which directly open the corresponding product series page in BGR's product center. The following four steps (with screenshots for the "*mb_lf*" product) explain how to find the download links of the data sets.

*Step 1: Landing page for the "mb_lf" DOI. Close the popup window to find the related data sets.*

*Step 2: The list of results in the background of the popup window (Fig. 1) shows the data series entry. Click on the icon "show datasets" to open a list of the data sets per year.*

*Step 3: Select the data set of interest by clicking on the title in the list. A new popup window will appear. (N.B.: the indicated icon directs you back to the data series entry).*

*Step 4: In the new popup window, the download link of the product's zip file can be found through the "Access" tab (line "Further information (Data)").*

We will check if we can add to the "Access" tab of each data series a link that directs to a list of all corresponding data sets. We will expand on this in line 656 of the manuscript in the final version.

Appendix Table A1. It would be helpful to include the year of installation or certification. This information is included in Figure 1, it should be included here too.

We added the "certification" column that provides the year of certification.

Table B1; lists data availability. It is not clear here what availability means. Presumably there are some gaps in data availability. As shown in Figure 1, not all stations are available for all 18 years. Does a product availability of 2.7 (for instance) mean that 2.7% that signals were present for 2.7% of the time that data were available? Or does it indicate that signals were present for 2.7% of the total 18 years. Some more description would be useful.

In general, an availability of 2.7% means that signals were present for 2.7% of the total 18 years (2003-2020). More precisely, a product is available for this portion of all time steps considered; e.g., with 30 min temporal resolution for the *maw* product, the number of time steps amounts to 315,600 over 18 years – 2.7% would be 8,521 time steps with non-NaN parameters available for a station. The data availability rate thus depends on multiple factors such as the operational time of a station, data gaps, the actual arrivals of coherent signals, and PMCC performance – to name the most relevant here.

We will better clarify the "data availability" in the table caption and Section 4.3.

Grammar/standard English usage:
There are some awkwardly worded sentences in this manuscript. Although they're

understandable, they sound awkward, and it would be helpful for a native English speaker to go over the manuscript carefully to catch them. I list a few below.

We will examine the document for such issues when incorporating all changes requested by both reviewers in the final version.

Line 12 and in the atmosphere --> or in the atmosphere

Fixed

Line 25 ...., each four products for 53 IMS infrasound stations were derived. Not sure what this means, does it mean ..., four products were derived for each of 53 IMS infrasound stations" ?

Indeed, "From the comprehensive detection lists, four products were derived for each of the 53 IMS infrasound stations."

Line 37: has been established --> was established

changed

Line 39: composing of ---> composed of

fixed

---

## Author Comment (AC2)

**Author reply to RC2 (essd-2021-441)**

Dear reviewer,

thank you very much for your comments, which helped improve our manuscript. We appreciate your positive recommendation and that you took the time to review our manuscript. We reply to your specific comments in detail below.

Yours sincerely,

Patrick Hupe on behalf of all co-authors
* * *
Comment: The abstract is quite lengthly and maybe not very focused on the core content of the paper itself. For example the part about improving meteorological models, volcanic explosion detection is tool long and would better suit in the introduction than in the abstract. Same about the fact that the IMS is supposed to detect 1 KT explosions, information about waveguides, etc. I would suggest to move all this in the Introduction. The key message in the abstract should probably be that a high-quality dataset is made available to the scientific community. Then the authors could briefly describe the dataset and then list all the possible/foreseen applications of this dataset.

Reply: Thanks for these remarks and suggestions. We have shortened the first part as suggested and added a few sentences at the end of the abstract. Changes are highlighted in the manuscript.

Line 12. "The IMS is supposed to detect any explosion of at least 1 kt of TNT equivalent underground, underwater, and in the atmosphere." -> There is no 1 kT minimum requirement within the CTBT. Maybe the sentence could be rephrase as "The IMS was initially designed to be able to detect any [...]" This would also be more in line with the "design goals" that are mentioned Line 121.

We rephrased this sentence and moved it to the introduction.

*"The IMS was initially designed to be able to detect and locate any nuclear explosion underground, underwater, or in the atmosphere."*

Line 40. I would remove the "respectively" as several technologies can detect a test in a same environment. Especially as later (Line 65), the authors provide examples of infrasound detection produced by underground explosions.

Accepted suggestion. We implemented this and split the long sentence into two sentences as follows:

*"The IMS was initially designed to be able to detect and locate any nuclear explosion underground, underwater, or in the atmosphere. When completed, this monitoring and verification infrastructure will consist of 337 facilities, composed of 170 seismic, 11 hydro-acoustic, and 60 infrasound stations. Eighty radionuclide stations and 16 radionuclide laboratories can provide evidence of the nuclear character of an explosion (e.g., Marty, 2019)."*

Line 41. I am not sure about the wording "radionuclide detector" vs. "stations" for waveform technologies. I would also suggest to use station for radionuclide technologies as a single station can include several "detectors". Or maybe use "facility".

We now use "stations", see previous comment.

Line 41. I would maybe add "16 radionuclide laboratories" as those only apply to this technology.

Good point. We inserted "radionuclide", see previous comments.

Line 59. "a flat response from 0.02 Hz to 4 Hz". The response is flat but +/-3 dB over this passband. This does not need to be added in the introduction but I did not find any information in the rest of the paper whether or not the amplitudes were corrected in the dataset from the response of the sensors used at the stations. This should mainly impact the lowest frequency bands. Something should be said about this in the paper.

We mention this in Section 2.2 now:

*"[…], accounting for a potentially lower resolution of the parameters at very low frequencies, being a result of the array response. The lower resolution may also apply to the amplitude parameters because the data were not corrected for the array response, which is considered flat between 0.02 and 4 Hz (±3 dB). Hereinafter a detection family is referred to as a detection."*

Line 130. IMS Stations are all referred to as "ISXX" in the paper. This has often been used in the past and is a minor comment only. Another option would be to use the official station names (as defined by ISC or CTBT) such as I01AR, etc. But that would probably be too much work as all data files are already named this way. Just a recommendation for the future.

With the "ISXX" we follow the IMS denotation, which seems a little easier to look up for non-IMS or non-CTBT user groups, because the "IXXNN" denotation is primarily used by the IDC and in its products (such as the REB, as listed in the ISC Bulletin). For this open-access data set, we deliberately chose to stick to the 4-digit denotation. It is also easier to handle, as not each country code needs to be known when using the data. However, since raw data of a couple of stations are also available through e.g. IRIS nodes (Incorporated Research

Institutions for Seismology), we have added a column to Table A1 showing the regional codes for reference.

*Table 1, caption: "The column 'code' denotes the country code as part of the 5-digit station codes (e.g., I26DE instead of IS26). We omit the 5-digit notation in our data products for simplicity, but it is common in IDC products and the vDEC."*

Line 133. "Station upgrades also lead to lacks of data since these often require a station to be revalidated." The revalidation process in most cases does not affect data availability. The upgrade process can (power off, etc.) but not the revalidation process. Station could be taken out of processing during the revalidation process (although not common anymore) but data availability would stay high.

We have clarified the two sentences:

*"Station upgrades, which are not depicted in Fig. 1b, can also lead to lacks of data (e.g., power temporarily off). Such upgrading activities include the installation of […]"*

Line 171. "The more sensors are progressively incorporated (generally from the inside to the outside of an array), the more potential aliasing is limited", I am not convinced that the limited aliasing is the main factor that allow improving the computation of wave parameters when the number of sensors increases for the selected processing technique (PMCC). Could the authors add some explanation here.

We agree that the limited aliasing is only one aspect. We have rephrased this sentence:

*"We use a consistency threshold of 0.1 s, which is centered in the range recommended by Runco Jr. et al. (2014). If more sensors are progressively incorporated (generally from the inside to the outside of an array), the precision of the estimated parameters is refined while initial false detections (e.g., sub-scale correlated noise) can be discarded; overall, the localization accuracy increases with increasing array aperture (Cansi, 1995; Cansi and Le Pichon, 2008)."*

Line 195: "Pixels adjacent to others in terms of time, back azimuth, and apparent velocity are grouped into detection families if at least 10 pixels contribute" -> not frequency ?

Indeed, also the frequency, thanks. We added "frequency" in this sentence and changed the following one accordingly:

*"The chosen  tolerance for the frequency criterion is a generous maximum of five bands, which is required due to the narrow bands at low frequencies. The maximum tolerances for the other parameters – 120 s, 10° to 5°, and 10 % to 5 % of the apparent velocity, respectively – are generally more constraining."*

Line 199-231: I would recommend to add what is the source of each of these artifact categories and how the applied criteria help filtering each of these artifacts. No explanation is given neither on how the thresholds such as the family size of 40 or 50 were chosen. The word "obvious" is used, but it is not obvious when reading the paper that a family size of 39 would be an artifact but 41 would definitely not be. So the chosen thresholds were probably defined based on statical analysis (ROC curves ?) and probably do not set a 100% clear line between "obvious artifacts" and real events. This also makes the sentence "We post-process the detection lists to discard obvious artefacts" quite strong statement as real events might have been filtered our as well.

The specific artefact sources are not resolved, but we state that PMCC seems sometimes unable to resolve the wave front parameters correctly, resulting in single-frequency-band detections. Of course, the threshold will always be a trade-off between true detection and false alarms rates. Also, due to this - and for different other reasons - we cannot guarantee the completeness of events in our detection lists. We clarified these issues in the paragraphs of Section 2.2.:

*"To our knowledge, standard thresholds for grouping pixels have not been specified, and threshold choices are a trade-off between the probability of detection and the false alarm rate (e.g., Runco Jr. et al., 2014)."*

*"We post-process the detection lists to discard the most obvious artefacts."*

*"In the same context and after considering 2D histograms of mean frequency and family size, we additionally clean the detection lists by discarding detections with family sizes <40; for frequencies of <0.06 Hz, detections with <50 pixels are discarded. The applied criteria are a trade-off between removing as many artefacts or false alarms and keeping as many actual events as possible."*

Line 228: "Effectively raising the lower family size threshold ensures the global comparability of the stations' detection lists and the derived products": Maybe not so clear for the reader. What derived products ?

We hopefully clarified this sentence:

*"Effectively raising the lower family size threshold from 10 to 40 and 50 ensures the global comparability of the stations' detection lists and any derived products in terms of this parameter, even though […]"*

No comments are made in Section 2 and 3 about the choice of the filter bank vs. the shape of the infrasound spectra. Were the spectra flattened before applying the PMCC processing ? If not, do we expect the frequency of the detection to be shifted towards higher detection because of leakage effects. Was some testing performed to compare the processing results with flatten spectra vs. raw data. Are the filters sharp enough ?

We did not explicitly compare the spectra, but since Hanning windows are implemented in the PMCC algorithm, the spectra should be flattened. We discuss this potential issue in Section 3.2 now, including references:

*"The centroids of these clusters seem to be shifted to slightly higher frequencies than before (by around 0.02 Hz). We do not observe such shifts for detection clusters at frequencies >0.1 Hz. However, compared to the low-noise models of selected IMS infrasound stations presented by Marty et al. (2021), the spectral (microbarom) peaks of Figs. 3 and 5 appear to be shifted to slightly higher frequencies by ~0.05 Hz. We cannot rule out some frequency uncertainty in the PMCC detections resulting from leakage effects due to the configuration settings. Garcés (2013) discussed potential energy leakage when assessing the benefit of using one-third octave bands for infrasound data processing. Hanning windows, which are implemented in PMCC for tapering the time windows and were also applied by Marty et al. (2021), should generally reduce potential energy leakage (e.g., Brachet et al., 2022). The shift in the frequency range <0.1 Hz could also result from better-constrained detection parameters using the one-third octave processing scheme."*

Line 310: "The white vertical lines near the center frequencies in (a) result from cleaning the detection list of ringing artefacts; with the newer version and configuration, the cleaning is easier to narrow down to the respective center frequencies." I think this is an interesting comment that only appears in the figure caption. This should probably be added to the text and a link made with the discussion at the end of Section 2.2.

We added this to Section 3.2, where it was indirectly said with the reference to Fig. 5. It is more explicitly written now:

*"Ringing artefacts at the respective frequency band centers were discarded at the expense of some true detections (white lines). Although the latter might also happen when using the newer version and configuration, the cleaning from artefacts seems more successfully applicable to the respective center frequencies, as the white lines of Fig. 5a almost disappear in Fig. 5b. Hence, based on the criteria explained in Section 2.2, discriminating between true detections and artefacts succeeds better with the 26-bands configuration. With this updated processing scheme,  we obtain […]"*

Section 3.3 (general comment as well): According to the authors, this dataset is mainly made available because the raw data is not available to the scientific community. But I think these dataset is also very valuable for those who have access to IMS data including the CTBTO. It allows identifying station performance issues that could be reported to the CTBTO and provide a very valuable dataset to all NDCs. Not all NDCs have the resources to compute such dataset and the fact that the authors made this available is great for everybody (not only those who don't have access). This is a positive point that should be emphasized more in the paper I think. I read it in the conclusion afterwards but it should probably be highlighted earlier in the paper.

Thank you for emphasizing this. We mention this earlier now.

Section 3.3, last sentence added: *"Overall, this parameter allows for identifying both naturally caused anomalies and station performance issues. Therefore, it is a valuable addition to the detection parameters of the data products and potentially of interest to station operators and NDCs, for instance, if they do not have the resources to compute such a data set."*

Section 5.4, added: *"If detected variations in the* mb *products cannot be associated with the atmospheric circulation pattern and its variability, they could also be another indication for station performance issues, along with the quality parameter (Section 3.3)."*

Line 369-376: There should probably be more explanation about why the microbarom band was divided into 2 categories. Because for the reader, it does not really appear very clearly in Figure 3 for example that there is 2 distinct categories in this frequency band. Maybe a some explanation could be added about the different use cases of these 2 datasets (and because this allows to have 2 different values instead of an averaged one over the entire frequency band).

We added the explanation before introducing the four products.

*"We decided to split the microbarom frequency range into two products because the peak frequency of the spectrum would generally dominate the detections within a single product. A second product potentially allows for better discriminating sources in this frequency range."^*

As a follow-up, we already refer to marginal seas and coastal regions in the discussion of Section 5.4.

Fig 8(b) the purple square (0.02-0.07 Hz) is shifted in time by about 20 min compared to the detection (does not align with the detection). This is probably because of the time is defined as the middle of a predefined time windows but something should probably be said about it.

We have checked this particular signature and the reason of this shift. We briefly explain this at the end of the first paragraph of Section 5.2 now:

*"Notably, the maw product (04:30 UTC) of Fig. 8b does not align with the first low-frequency PMCC detections before 04:00 UTC. However, these are broadband detection families whose mean frequencies are higher and therefore not assigned to the maw product. Within the 30 min after 04:00 UTC, a few low-frequency detections result in the maw product signature of 04:30 UTC."*

Figure 9. The legend about what the color-coded squares are is missing.

The legends of Fig. 8 apply to Fig. 9, too. We state this in the last sentence of the caption.

Line 681-684: This is true and should be included in the paper, but I am not sure it is one of the main highlights of the paper that should be included in a conclusion. As for the abstract itself, the conclusion could be slightly re-written to better summarize the main points of the paper with possible opening (future work).

We slightly rephrased the conclusion to address this comment and better highlight the main points. Thank you!

---

## Author Response (AR2)

**Author reply during 2ⁿᵈ revision (essd-2021-441)**

Dear reviewers,

we appreciate that you took the time to review our manuscript a second time. We acknowledge your reviewing efforts in the Acknowledgements section. Also, thank you very much for your positive recommendation again.
As was anticipated in the initial and revised manuscript versions, the "Produktcenter" of BGR is now being replaced by the "Geoportal", leading to better functionality and easier-to-find download links. To address your comments from the second review in the revised version, we decided to refer to the new Geoportal only (as it is now operational) and, in addition, to simplify the download of the data sets further. The latter is achieved by desisting from the 'data series' with subordinary data sets (one per year and product). Instead, each data product can now be downloaded as a single data set (.zip file for one product containing all years and stations). Consequently, the DOI of a product directly points to the respective data set in the data portal rather than a data series. There are no hard-to-find related data sets anymore, and a download link is provided directly on the landing page or next to the search result in the Geoportal. Since all years are compressed within one zip file, the download file sizes have considerably increased. But overall, the products have become much more accessible. We refer to the details given in the response below.

Yours sincerely,

Patrick Hupe on behalf of all co-authors
* * *
Comment (Catherine de Groot-Hedlin): Thank you for the responses to my initial review. The manuscript is clearer now and the added information is Tables A1 and B1 is helpful. I still have doubts about the ease of data access. Thank you for including detailed information in your response regarding how to access the data. I have been able to confirm that the data are indeed available. However, the manuscript does not include these detailed steps on how to access the data, which may be confusing as there are several windows to click through; clicking on the given links does not lead to immediate access. It would be helpful to go into a bit more detail on what steps to take. I am not suggesting including the screenshots of all the steps, but maybe list the steps, i.e. a) clear the popup window, b) click on the 'show datasets' button, c) click on the title of a dataset in the new window that pops up, d) click on access, etc. I was able to find the data using the steps you suggested starting from the DOI links but was still unable to find it using the BGR link. Please consider either removing the BGR link or else include further explanations on how to get the data that way.

Again, I recommend that publication of this manuscript after consideration of this minor edit.

Reply: Thank you for your feedback on the ease of data access, which is much appreciated. As mentioned in the cover letter above, the DOIs link to the new Geoportal now, superseding the formerly used product center. The 'data series plus data set'-structure has been replaced by a single data set per product. We believe that these two changes make it much easier to find and download the zipped data files. We have accordingly adapted Section 6 of the manuscript. The data access description reduces to the following:

*The infrasound data products are openly accessible through the product center ('Geoportal', https://geoportal.bgr.de/, last accessed on 26 July 2022) of the Federal Institute for Geosciences and Natural Resources (BGR), the German NDC (Pilger et al., 2017). For each of the four product types, the Geoportal contains one data set that is assigned a DOI (Table 2). Each data set is provided as a compressed .zip file. The .zip files contain a README file, yearly subdirectories with the netCDF (.nc) data files for all certified stations, and a simple MATLAB code that reads and plots the netCDF data of a station.*

*With the DOI as the search item, a data product can be found in the Geoportal using the search function. Alternatively, search 'infrasound' and set the filter 'dataset' for displaying a list of all available infrasound data products (i.e., not limited to these four data sets). The direct landing page of a DOI (e.g., https://doi.org/10.25928/bgrseis_bblf-ifsd for 'maw') opens the metadata page of the respective product (this page may take a few seconds to load). The download link of a data set (.zip file) is displayed on the right side of both the search result and the metadata page ('Download – Select type – Download-Link').*

*Table 1: DOIs and references related to the infrasound products. The file sizes refer to the 2003–2020 data sets and will increase when more recent years are added to the .zip files.*

| Product | DOI | Reference | Download size (.zip) |
|---------|-----|-----------|----------------------|
| *maw* | *10.25928/bgrseis_bblf-ifsd* | *Hupe et al., 2021a* | *~100 MB* |
| *mb_lf* | *10.25928/bgrseis_mblf-ifsd* | *Hupe et al., 2021b* | *~900 MB* |
| *mb_hf* | *10.25928/bgrseis_mbhf-ifsd* | *Hupe et al., 2021c* | *~320 MB* |
| *hf* | *10.25928/bgrseis_bbhf-ifsd* | *Hupe et al., 2021d* | *~1.1 GB* |